# Towards mental time travel: a hierarchical memory for reinforcement learning agents

**Andrew K. Lampinen**
DeepMind
London, UK
lampinen@deepmind.com

**Stephanie C. Y. Chan**
DeepMind
London, UK
scychan@deepmind.com

**Andrea Banino**
DeepMind
London, UK
abanino@deepmind.com

**Felix Hill**
DeepMind
London, UK
felixhill@deepmind.com

## Abstract

Reinforcement learning agents often forget details of the past, especially after delays or distractor tasks. Agents with common memory architectures struggle to recall and integrate across multiple timesteps of a past event, or even to recall the details of a single timestep that is followed by distractor tasks. To address these limitations, we propose a Hierarchical Chunk Attention Memory (HCAM), which helps agents to remember the past in detail. HCAM stores memories by dividing the past into chunks, and recalls by first performing high-level attention over coarse summaries of the chunks, and then performing detailed attention within only the most relevant chunks. An agent with HCAM can therefore "mentally time-travel"—remember past events in detail without attending to all intervening events. We show that agents with HCAM substantially outperform agents with other memory architectures at tasks requiring long-term recall, retention, or reasoning over memory. These include recalling where an object is hidden in a 3D environment, rapidly learning to navigate efficiently in a new neighborhood, and rapidly learning and retaining new object names. Agents with HCAM can extrapolate to task sequences an order of magnitude longer than they were trained on, and can even generalize zero-shot from a meta-learning setting to maintaining knowledge across episodes. HCAM improves agent sample efficiency, generalization, and generality (by solving tasks that previously required specialized architectures). Our work is a step towards agents that can learn, interact, and adapt in complex and temporally-extended environments.

## 1 Introduction

Human learning and generalization relies on our detailed memory of the past [55, 52, 16, 30, 43]. If we watch a show, we can generally recall its scenes in some detail afterward. If we explore a new neighborhood, we can recall our paths through it in order to plan new routes. If we are exposed to a new noun, we can find that object by name later. We experience relatively little interference from delays or intervening events. Indeed, human episodic memory has been compared to "mental time travel" [55, 52, 37]—we are able to *transport* ourselves into a past event and re-live it in sequential detail, without attending to everything that has happened since. That is, our recall is both sparse (we attend to a small chunk of the past, or a few chunks) *and* detailed (we recover a large amount of information

35th Conference on Neural Information Processing Systems (NeurIPS 2021).

from each chunk). This combination gives humans the flexibility necessary to recover information from memory that is specifically relevant to the task at hand, even after long periods of time.

If we want Reinforcement Learning (RL) agents to meaningfully interact with complex environments over time, our agents will need to achieve this type of memory. They will need to remember event details. They will need to retain information they have acquired, despite unrelated intervening tasks. They will need to rapidly learn many pieces of new knowledge without discarding previous ones. They will need to reason over their memories to generalize beyond their training experience.

However, current models struggle to achieve rapid encoding combined with detailed recall, even over relatively short timescales. Meta-learning approaches [57] can slowly learn global task knowledge in order to rapidly learn new information, but they generally discard the details of that new information immediately to solve the next task. While LSTMs [20], Transformers [56], and variants like the TransformerXL [9] can serve as RL agent memories in short tasks [44], we show that they are ineffective at detailed recall even after a few minutes. Transformer attention can be ineffective at long ranges even in supervised tasks [54], and this problem is likely exacerbated by the sparse learning signals in RL. By contrast, prior episodic memory architectures [21, 14] can maintain information over slightly longer timescales, but they struggle with tasks that require recalling a temporally-structured event, rather than simply recalling a single past state. That is, they are particularly poor at the type of event recall that is fundamental to human memory. We suggest that this is because prior methods lack 1) memory chunks longer than a single timestep and 2) sparsity of attention. This means that these models cannot effectively "time-travel" to an event, and relive that specific event in detail, without interference from all other events. This also limits their ability to reason over their memories to adapt to new tasks—for example, planning a new path by combining pieces of previous ones [49].

Here, we propose a new type of memory that begins to address these challenges, by leveraging sparsity, chunking, and hierarchical attention. We refer to our architecture as a Hierarchical Chunk Attention Memory (HCAM). The fundamental insight of HCAM is to divide the past into distinct chunks before storing it in memory, and to recall hierarchically. To recall, HCAM first uses coarse chunk summaries to identify relevant chunks, and then mentally travels back to attend to each relevant chunk in further detail. This approach combines benefits of the sparse and relatively long-term recall ability of prior episodic memories [60, 49] with the short-term sequential and reasoning power of transformers [56, 9, 44] in order to achieve better recall and reasoning over memory than either prior approach. While we do not address the problem of optimally chunking the past here, instead employing arbitrary chunks of fixed length, our results show that even fixed chunking can substantially improve memory. (Relatedly, Ke et al. [25] have shown that sparse, top-$k$ retrieval of memory chunks can improve credit assignment in LSTMs with attention.)

We show that HCAM allows RL agents to recall events over longer intervals and with greater sample efficiency than prior memories. Agents with HCAM can learn new words and then maintain them over distractor phases. They can extrapolate far beyond the training distribution, to maintain and recall memories after $5\times$ more distractors than during training, or after multiple episodes when trained only on single ones. Agents with HCAM can reason over their memories to plan paths near-optimally as they explore a new neighborhood, comparable to an agent architecture designed specifically for that task. HCAM is robust to hyperparameters, and agents with HCAM consistently outperform agents with Gated TransformerXL memories [44] that are twice as wide or twice as deep. Furthermore, HCAM is more robust to varying hyperparameters than other architectures (App. D.11). Thus HCAM provides a substantial improvement in the memory capabilities of RL agents.

## 1.1 Background

**Memory in RL**   Learning signals are sparse in RL, which can make it challenging to learn tasks requiring long memory. Thus, a variety of sophisticated memory architectures have been proposed for RL agents. Most memories are based on LSTMs [20], often with additional key-value episodic memories [60, 14]. Often, RL agents require self-supervised auxiliary training because sparse task rewards do not provide enough signal for the agent to learn what to store in memory [60, 14, 19].

**Transformers**   Transformers [56] are sequence models that use attention rather than recurrence. These models—and their variants [9, 28, 58, 46]—have come to dominate natural language processing. This success of transformers over LSTMs on sequential language tasks suggests that they might be effective agent memories, and a recent work succesfully used transformers as RL agent memories

[44]—specifically, a gated version of TransformerXL [9]. However, using TransformerXL memories on challenging memory tasks still requires auxiliary self-supervision [19], and might even benefit from new unsupervised learning mechanisms [3]. Furthermore, even in supervised tasks Transformers can struggle to recall details from long sequences [54]. In the next section, we propose a new hierarchical attention memory architecture for RL agents that can help overcome these challenges.

## 2 Hierarchical Chunk Attention Memory

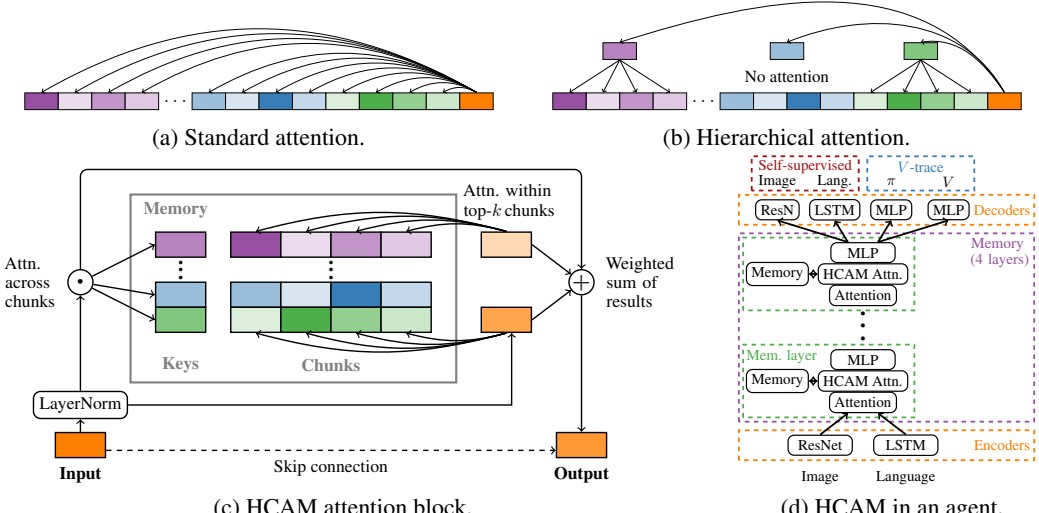

Figure 1: HCAM motivation and implementation. (a) Standard transformer attention attends to every time step in a fixed window. (b) HCAM divides the past into chunks. It first attends to summaries of each chunk, and only attends in greater detail within relevant chunks. (c) HCAM is implemented using a key-value memory, where the keys are chunk summaries, and the values are chunk sequences. Top-level attention is performed over all summaries (left), and then standard attention is performed within the top-$k$ chunks. The results are relevance-weighted and summed (right), and then added to a residual input to produce the block output. (d) Our agent encodes inputs, processes them through a 4-layer memory with an HCAM block in each layer, and decodes outputs for self-supervised and RL losses.

The HCAM uses a hierarchical strategy to attend over memory (Fig. 1). Specifically, it divides the past into chunks. It stores each chunk in detail—as a sequence—together with a single summary key for that chunk. It can then recall hierarchically, by first attending to chunk summaries to choose the top-$k$ relevant chunks, and then attending within those most relevant chunks in further detail. In particular, let $S$ denote the memory summary keys and $Q$ be a linear projection ("query") layer. Then the input is normalized by a LayerNorm, and the chunk relevance ($R$) is computed as:

$$R = \text{softmax}(Q(\text{normed input}) \cdot S)$$

Then the model attends in detail within the chunks that have the top-$k$ relevance scores. If the memory chunks are denoted by $\mathcal{C}$ and MHA(query inputs, key/value inputs) denotes multi-head attention:

$$\text{memory query results} = \sum_{i \in \text{top-}k \text{ from } R} R_i \cdot \text{MHA}(\text{normed input}, \mathcal{C}_i)$$

That is, HCAM selects the most relevant memory chunks, then time travels back into each of those chunks to query it in further detail, and finally aggregates the results weighted by relevance. This result is added to the input to produce the output of the block. An HCAM block can be *added* to a transformer layer, to supplement standard attention. (For an open-source HCAM implementation, see App. A.1)

HCAM can attend to any point in its memory, but at each step it only attends to a few chunks, which is much more compute- and memory-efficient than full attention. With $N$ chunks in memory, each of size $C$, using HCAM over the top-$k$ chunks requires $O(N + kC)$ attention computations, whereas full attention requires $O(NC)$. HCAM succeeds even with $k = 1$ in many cases (App. D.5), reducing computation $10\times$ compared to TrXL. Even with $k \geq 1$, HCAM often runs faster (App. D.10).

**Agents** Our agents (Fig. 1d) consist of input encoders (a CNN or ResNet for visual input, and an LSTM for language), a 4-layer memory (one of HCAM, a Gated TransformerXL [44], or an LSTM [20]), followed by output MLPs for policy, value, and auxiliary losses. Our agents are trained using IMPALA [11]. To use HCAM in the memory, we adapt the Gated TransformerXL agent memory [44]. We replace the XL memory with an HCAM block between the local attention (which provides short-term memory) and feed-forward blocks of each layer. We find that gating each layer [44] is not beneficial to HCAM on our tasks (App. D.7), so we removed it. See App. A.2 for further agent details.

We store the *initial layer inputs* in memory—that is, the inputs to the local attention—which performs better than storing the output of the local attention, and matches TransformerXL (TrXL) more closely. We accumulate these inputs into fixed-length memory chunks. When enough steps have accumulated to fill the current chunk, we add the current chunk to the memory. We average-pool across the chunk to generate a summary. We reset the memory between episodes, except where cross-episode memory is tested (Section 3.3). We stop gradients from flowing into the memory, just as TrXL [9] stops gradients flowing before the current window. This reduces challenges with the non-differentiability of top-$k$ selection (see App. D.5 for discussion). It also means that we only need to store past activations in memory, and not the preceding computations, so we can efficiently store a relatively long time period.

However, the model must therefore learn to encode over shorter timescales, since it cannot update the encoders through the memory. Thus, as in prior work referenced above, we show that *self-supervised learning* [33], is necessary for successful learning on our tasks (App. D.6). Specifically, we use an auxiliary *reconstruction* loss—at each step we trained the agents (of all memory types) to reconstruct the input image and language as outputs [19]. This forces the agents to propagate these task-relevant features through memory, which ensures that they be encoded [cf. 21, 14].

## 3 Experiments

We evaluated HCAM in a wide variety of domains (Fig. 2). Our tasks test the ability of HCAM to recall event details (Sec. 3.1); to maintain object permanence (Sec. 3.2); to rapidly learn and recall new words, and extrapolate far beyond the training distribution (Sec. 3.3); and to reason over multiple memories to generalize, even in settings that previously required task-specific architectures (Sec. 3.4).

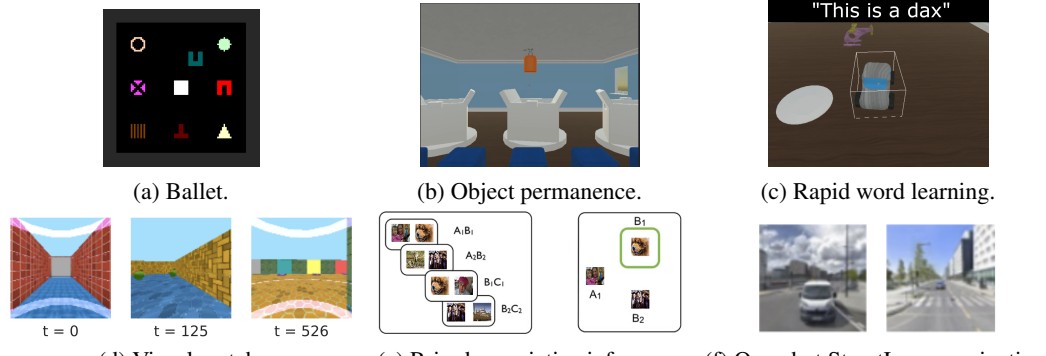

Figure 2: We evaluated our model across a wide variety of task domains, some of which are modified or used from other recent memory papers [19, 21, 2, 49]. These include a variety of environments (2D, 3D, real images) and tasks (finding hidden objects, language learning, and navigation).

### 3.1 Remembering the ballet

Our first experiment tests the ability to recall spatiotemporal detail, which is important in real-world events. We place the agent in a 2D world, surrounded by "dancers" with different shapes and colors (Fig. 2a). One at a time, each dancer performs a 16-step solo dance (moves in a distinctive pattern), with a 16- or 48-step delay between dances. After all dances, the agent is cued to e.g. "go to the dancer who danced a square," and is rewarded if it goes to the correct dancer. This task requires recalling dances in detail, because no single step of a dance is sufficient to distinguish it from other dances. The task difficulty depends on the number of dances the agent must watch in an episode (between 2 and 8) and the delay length. We varied both factors across episodes during training—this

likely increases the memory challenges, since the agent must adapt in different episodes to recalling different amounts of information for varying periods. See App. B.2 for details.

Fig. 3 shows the results. Agents with an HCAM or a Gated TransformerXL (TrXL) memory perform well at the shortest task, with only 2 dances per episode (Fig. 3a). LSTM agents learn much more slowly, showing the benefits of attention-based memories. With 8 dances per episode (Fig. 3b), HCAM continues to perform well. However, TrXL's performance degrades, **even though the entire task is short enough to fit within a single TrXL attention window**. (The dances are largely in the XL region, where gradients are stopped for TrXL, but HCAM has the same limitation.) LSTM performance also degrades substantially. The advantage of HCAM is even more apparent with longer delays between the dances (Fig. 3c). HCAM can also generalize well ($\geq 90\%$) from training on up to 6 dances to testing on 8 (App. D.2). Furthermore, HCAM is robust to varying memory chunk sizes (App. D.4), even if the chunk segmentation is not task aligned. Finally, HCAM can perform well on this task even if it is only allowed to select a single memory chunk rather than the top-$k$ (App. D.5), thereby attending to $8\times$ fewer time-points than TrXL (while a sparse TrXL is not advantageous, see App. D.9). Thus HCAM can increase computational efficiency. In summary, both attention-based memories recall events better than LSTMs, and HCAM robustly outperforms TrXL at all but the easiest tasks.

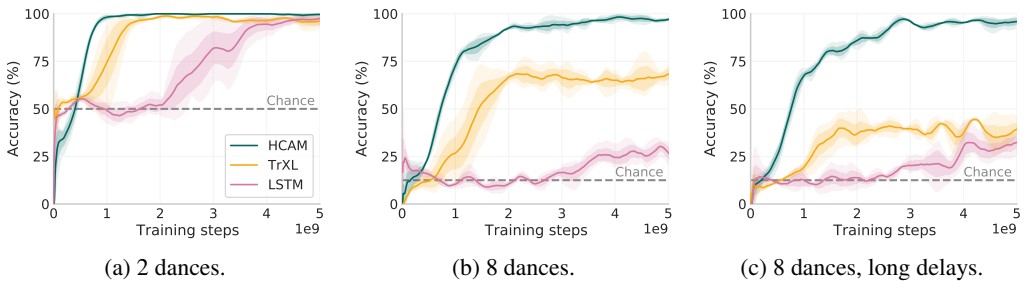

(a) 2 dances.      (b) 8 dances.      (c) 8 dances, long delays.

Figure 3: The ballet task—HCAM (teal) outperforms TrXL (yellow) and LSTM (red) memories at difficult ballets with many dances, especially when there are long delays between. (a) In the shortest ballets, TrXL performs nearly as well as HCAM, while LSTMs require much more training. (b) With 8 dances, HCAM substantially outperforms TrXL, even though the entire task fits within one TrXL attention window. LSTMs do not get far above chance at choosing among the dancers. (c) With longer delays between dances, HCAM performs similarly but TrXL degrades further. (3 runs per condition, lines=means, light regions=range, dark=$\pm$SD. Training steps are agent/environment steps, not the number of parameter updates. Chance denotes random choice among the dancers, not random actions.)

## 3.2 Object permanence

Our next tasks test the ability to remember the identity and location of objects after they have been hidden within visually-identical boxes, despite delays and looking away. These tasks are inspired by work on object permanence in developmental psychology [e.g. 1]. We implemented these tasks in a 3D Unity environment. The agent is placed in a room and fixed in a position where it can see three boxes (Fig. 4a). One at a time, an object jumps out of each box a few times, and then returns to concealment inside. There may be a delay between one object and the next. After all objects have appeared, the box lids close. Then, the agent is released, and must face backwards. Finally, it is asked to go to the box with a particular object, and rewarded if it succeeds.

HCAM solves these tasks more effectively than the baselines. HCAM learns the shortest tasks (with no delays) substantially faster than TrXL (Fig. 4b), and LSTMs are unable to learn at all. When trained on tasks of varying lengths, HCAM was also able to master tasks with 30 seconds of delay after each object revealed itself, while TrXL performed near chance even with twice as much training (Fig, 4b). Furthermore, HCAM can learn these long tasks even without the shorter tasks to scaffold it (Fig. 4d).

## 3.3 Rapid word learning with distractor tasks & generalization across episodes

The preceding tasks focused on passively absorbing events, and then recalling them after delays. We next turn to a set of rapid-word-learning tasks (Fig. 2c) that require actively acquiring knowledge, and maintaining it in more challenging settings, with intervening distractor tasks (Fig: 5a). These

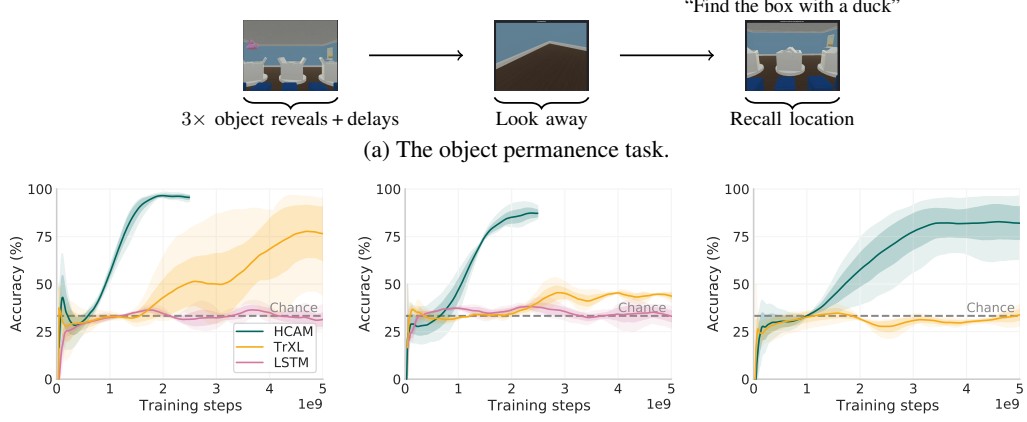

(a) The object permanence task.

(b) No delays, varying training.  (c) Long delays, varying training.  (d) Long delays, long-only training.

Figure 4: The object permanence task—HCAM learns faster than TrXL, and succeeds at longer tasks. (a) The task structure. (b) The shortest tasks, with no delays. When trained with varying delay lengths, HCAM rapidly and consistently learns to track object locations. TrXL learns more slowly and less consistently. (b) Long tasks, with 30-second delays between each object being revealed. HCAM learns quite well, while TrXL barely achieves above-chance performance. (d) HCAM can learn the long tasks even without simultaneous training on the shorter tasks to scaffold its learning. (3 runs per condition. Chance denotes random choice among the boxes, not random actions.)

tasks are based on the work of Hill et al. [19], who showed that transformer-based agents in a 3D environment can meta-learn to bind objects to their names after learning each name only once. That is, immediately after a single exposure to novel object names, their agent can successfully lift objects specified by those names. Their agent then discards its memory before proceeding to the next episode. However, in more complex environments tasks will not be so cleanly structured. There will often be distracting events and delays—or even learning of other information—before the agent is tested.

We therefore created more challenging versions of these tasks by inserting distractor tasks between the learning and test phases. In each distractor task we placed the agent in a room with a fixed set of three objects (that were never used for the word-learning portion), and asked it to lift one of those objects (using fixed names). By varying the number of distractor phases, we were able to manipulate the demands on the memory. We trained our agents on tasks with 0, 1, or 2 distractor phases, and evaluated their ability to generalize to tasks with more distractors. HCAM and TrXL could learn the training tasks equally well, and both showed some ability to generalize to longer tasks (App. D.1). However, HCAM was able extrapolate better to tasks with 10 distractor phases (Fig. 5d), and even showed some extrapolation to 20 distractors, an order of magnitude more than trained (Fig. 7a).

**Maintaining knowledge across episodes zero-shot**    It is a fundamental limitation of most meta-learning paradigms that they discard the information they have learned after each episode. The agents used by Hill et al. [19] discarded their memories end of each episode. Children, by contrast, can rapidly learn new words and maintain that knowledge even after learning other new words. This leads to a question: if HCAM can effectively maintain a word-object mapping despite intervening distractor tasks, could it maintain this knowledge across entire episodes of learning new words?

To answer this question, we used the agents trained on single-episode tasks. We evaluated these agents on tasks where they were sometimes tested on a word from several episodes before, despite having learned and been tested on other words in the intervening period (Fig. 5b). In training, the agents were never required to maintain a set of words once they encountered a new learning phase with new words. Nevertheless, our agent with HCAM is able to generalize well to recalling words four episodes later (Fig. 5e). It can even extrapolate along multiple dimensions, recalling words two episodes later with more distractor phases per episode than it ever saw in training (Fig. 5f). In App. D.3 we show how the attention patterns of the hierarchical memory support this generalization across episodes. **Agents with HCAM are able to generalize to memory tasks that are far from the training distribution.**

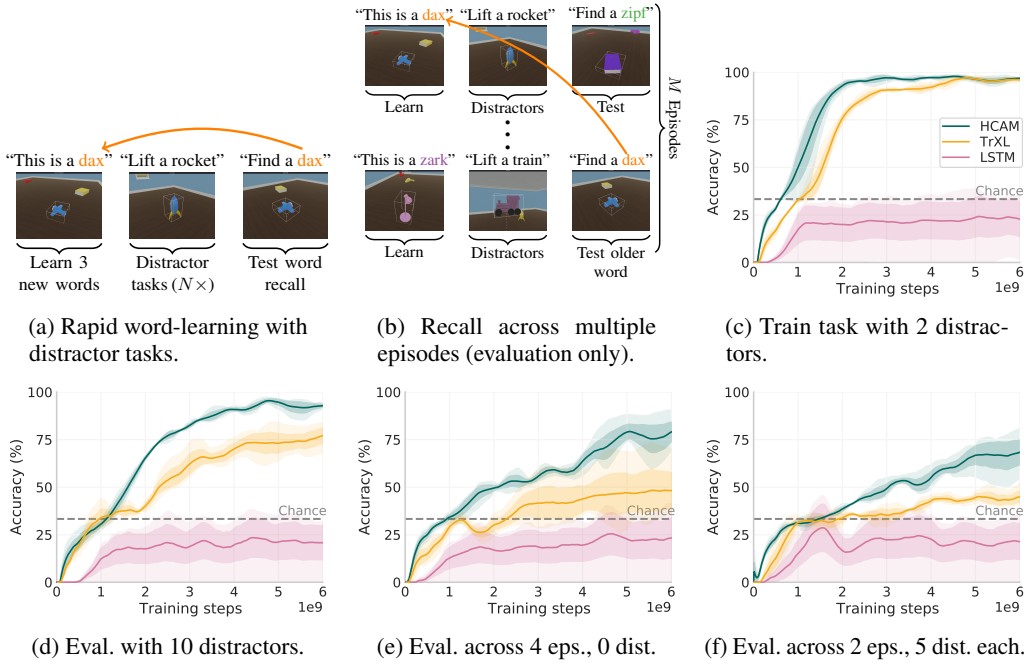

(a) Rapid word-learning with distractor tasks.

(b) Recall across multiple episodes (evaluation only).

(c) Train task with 2 distractors.

(d) Eval. with 10 distractors.

(e) Eval. across 4 eps., 0 dist.

(f) Eval. across 2 eps., 5 dist. each.

Figure 5: The rapid word-learning tasks. (a) Training tasks consist of learning three words (nouns), completing distractor tasks, and then finding an object named using one of the three learned words. We evaluated generalization to larger numbers of distractors. (b) During training, the agent never had to recall words for more than a single episode. However, we also evaluated the agent on its ability to recall a word learned several episodes earlier. This tests the ability of our memory to bridge from rapid meta-learning to longer-term retention. (c) Agents with HCAM or TrXL memories learn the hardest training task (2 distractors) well, although the agent with HCAM is faster to learn. When trained only on single episodes with 0-2 distractor phases: (d) HCAM outperforms TrXL at extrapolation to 10 distractor phases. (e-f) HCAM can generalize strongly out-of-distribution to recalling words that were learned several episodes earlier, despite having completed other episodes in the intervening time. (3 runs per condition. Chance denotes random performance on the final evaluation choice; this requires complex behaviours: learning all names and completing all distractors and intermediate episodes to reach the final choice. This is why LSTM performance is below chance in one seed.)

## 3.4 Comparisons with alternative memory systems

There has been an increasing interest in architectures for memory in RL, as well as in deep learning more broadly. In order to help situate our approach within this literature, we benchmarked our model against tasks used in several recent papers exploring alternative forms of memory. HCAM is able to perform well across all the tasks—reaching near-optimal performance, comparable with memory architectures that were specifically engineered for each task—and outperforms strong baselines.

**Passive Visual Match** First, we compare to the Passive Visual Match task from Hung et al. [21]. In this task, the agent must remember a color it sees at the beginning of the episode, in order to choose a matching color at the end (Fig. 2d). HCAM achieves reliably accurate performance (Fig. 6a), comparable to RMA, the best model from Hung et al. [21]. It would be interesting to combine HCAM with the value transport idea Hung et al. proposed, in order to solve their active tasks.

**Paired Associative Inference** We next considered the Paired Associative Inference task [2]. In each episode, the agent receives a set of pairings of stimuli, and it must chain together transitive inferences over these pairings in order to respond to a probe (Fig. 2e). These tasks therefore require sequential reasoning over multiple memories. HCAM achieves comparable performance to MEMO on the length 3 version of this task, and substantially outperforms baselines like Universal Transformer [10]. This is likely because HCAM, like MEMO, respects the chunk structure of experiencing pairs of images.

**One-Shot StreetLearn** Finally, we evaluated HCAM on One-Shot StreetLearn (Fig. 2f) proposed by Ritter et al. [49], based on Mirowski et al. [40]. One-shot StreetLearn is a challenging meta-

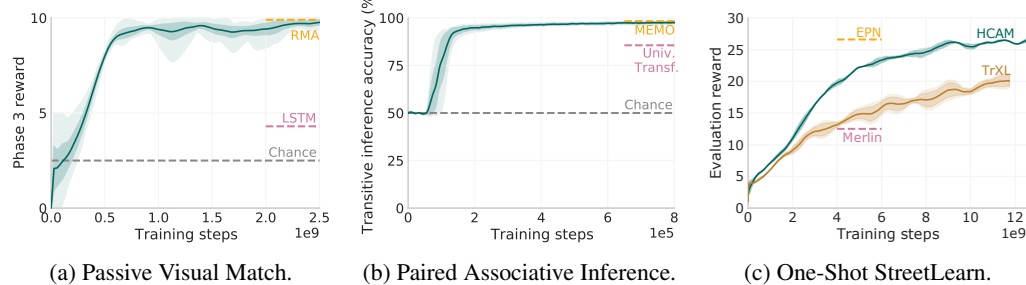

| (a) Passive Visual Match. | (b) Paired Associative Inference. | (c) One-Shot StreetLearn. |

Figure 6: Evaluating HCAM on tasks from other memory papers. Our agent achieves performance competitive with the specialized model proposed to solve each task. Top results (gold) and selected baselines (red) from each paper are indicated. (a) The Passive Visual Match task [21]. (a) The Paired Associative Inference task [2]. HCAM achieves near optimal performance, comparable to MEMO and above Universal Transformers [10]. (c) The One-Shot StreetLearn environment [49]. HCAM achieves comparable performance to EPN on this difficult navigation task, although it is slower to learn. Both substantially outperform strong baselines. (Some prior results were estimated from figures. HCAM results aggregate over: (a) 6 seeds. (b) 3 seeds, selected by validation accuracy. (c) 2 seeds.)

learning setting. The agent is placed in a neighborhood, and must navigate to as many goals as it can within a fixed time. The agent receives its current state and goal as visual inputs (Google StreetView images) and must learn about the neighborhood in early tasks within an episode, in order to navigate efficiently later. Ritter et al. designed a specialized Episodic Planning Network (EPN) to solve these tasks. HCAM—despite its more general purpose architecture—can achieve comparable performance to EPN (although HCAM is somewhat slower to reach this level of performance). Ritter et al. showed that EPN achieves near-optimal planning in later parts of each episode, and therefore **HCAM must be planning close to optimally to match EPN's performance.** Strong baselines perform much worse. Merlin [60]—a strong agent with a learned episodic key-value memory and auxiliary unsupervised training—only achieves around half the performance of HCAM and EPN. TrXL is also less effective. This highlights the value of more sophisticated memory architectures, and in particular emphasizes the value of sequences stored in memory, rather than single vectors: EPN stores transitions (sequences of length 2) and HCAM stores longer sequences, while Merlin has only single vector values for each key (as does TrXL).

## 4 Discussion

In this paper we have proposed the Hierarchical Chunk Attention Memory. HCAM allows agents to attend in detail to past events without attending to irrelevant intervening information. Thus, HCAM effectively implements a form of "mental time-travel" [55]. Our main insight is that this ability can be achieved using a hierarchically structured memory that sparsely distributes detailed attention. Specifically, recall uses attention over memory chunk summaries to allocate more detailed attention within only the most relevant chunks. We see this as a potentially important insight, because both mental time travel [52] and hierarchical structure [16] are essential to the power of human memory. Furthermore, recent work has suggested that human memory behaves more like a "quantum wave function" distribution over several past events [37], which matches our approach of a distribution over the top-$k$ relevant chunks. HCAM incorporates a number of important features of human memory.

Correspondingly, we showed that agents with HCAM succeed at a wide range of environments and tasks that humans depend on memory to solve. HCAM allows agents to remember sequential events, such as a ballet, in detail. HCAM allows agents to rapidly learn to navigate near-optimally in a new neighborhood by planning over previous paths; to perform transitive inferences; and to maintain object permanence, despite delays, occlusion and looking away. HCAM allows agents to effectively learn new words from a single exposure, and maintain that knowledge across distractor tasks. The agents can even extrapolate far beyond the training distribution to recall words after subsequent learning episodes **without ever being trained to do so**—better memory systems can help with the challenge of bridging from meta-learning to continual learning, which is receiving increasing interest [13, 17, 49, 5, 41].

The tasks we considered are challenging because they rely on flexible use of memory. The agent did not know in advance which paths to remember in One-Shot StreetLearn, or which dancer would

be chosen in Ballet. Furthermore, we trained the agent simultaneously on procedurally generated episodes where task features varied—e.g. the agent did not know how many distractor phases would appear in an episode, or the length of delays, so it could not rely on a fixed policy that recalls information after a fixed time. HCAM might be especially useful in these difficult, variable settings.

HCAM is robust to hyperparameters such as chunk size (App. D.4) and the number $k$ of memories selected at each layer and step (App. D.5)—it is even able to solve many tasks with $k = 1$. It also outperforms TrXL models that are either twice as wide or twice as deep, and correspondingly have many more parameters and, in the deeper case, make many more attention computations (App. D.8). Our comparisons to other approaches (Sec. 3.4) show that our approach is competitive with state of the art, problem-specific memory architectures, and outperforms strong baselines like Merlin and Universal Transformers. Finally, in hyperparameter sweeps suggested by our reviewers to improve TrXL's performance, we found that HCAM was consistently more robust to hyperparameter variation (App. D.11). These observations suggest that our results should be relatively generalizable.

**Self-supervised learning** As in prior memory work [60, 14, 19] we found that self-supervised learning was necessary to train the agent to store all task-relevant information in memory. Training the agent to reconstruct its input observations as outputs was sufficient in the tasks we explored, but more sophisticated forms of auxiliary learning [23, 3] might be useful in other settings.

**Memory in RL vs. supervised settings** Our work has focused on improving the memory of a situated RL agent. Compared to supervised settings such as language modelling, where a vanilla transformer can achieve very impressive performance [48, 4], RL poses unique challenges to memory architectures. First, sparse rewards present a more impoverished learning signal than settings that provide detailed errors for each output. The ability of HCAM to restore a memory in detail may help ameliorate this problem. Second, the multimodal (language + vision) stimuli experienced by our agent contains much more information per step than the word-pieces alone that a language model experiences, and therefore our setting may place more demands on the memory architecture. Finally, our tasks require access to detailed, structural aspects of past memories, while many existing NLP tasks do not depend significantly on structure — for example, Pham et al. [45] show that BERT ignores word order when solving many language understanding benchmarks, and produces equivalent outputs for shuffled inputs. Detailed memory will be most beneficial in settings in which the sequential structure of the past is important.

Nonetheless, we do not rule out possible applications of HCAM in supervised settings, such as language processing [48, 4], video understanding [53], or multimodal perception [24]. HCAM would likely be most beneficial in tasks that strongly rely on long-term context and structure, e.g. the full-length version of NarrativeQA [29]. Because videos often contain hierarchically-structured events, video models might especially benefit from HCAM-style attention. More broadly, the greater efficiency of sparse, hierarchical attention might allow detailed memory even in settings with limited computational resources, such as embedded systems.

**Episodic memories** Nematzadeh et al. [42] suggested endowing transformers with external episodic memories. Several recent papers have proposed systems that can be interpreted as episodic memories from this perspective, specifically looking up related contexts from the training corpus using either nearest neighbors [26], or attention [62]. However, these approaches have generally only used this type of external memory to predict outputs, rather than allowing the model to perform further computations over the memories it recalls, as in HCAM. Thus, these memories would be inadequate for most RL tasks, which require planning or reasoning with memories.

Various works have proposed other types of external/episodic memory, both in RL specifically [e.g. 60, 21, 14] and in deep learning more broadly [e.g. 50, 15]. These approaches have generally not stored memories with the hierarchical summary-chunk structure of HCAM. Ritter et al. [49] proposed an episodic memory for navigation tasks that stores state transitions; this architecture can be seen as a step towards our approach of storing sequences as chunks. Indeed, in the challenging city navigation domain that Ritter et al. explored, we showed that HCAM achieves comparable performance to their EPN architecture, despite HCAM being much more general. Furthermore, both HCAM and EPN substantially outperform Merlin [60], a strong baseline that learns to store vector memories, rather than the rich representations of sequences stored by HCAM (or transitions stored by EPN). This highlights the value of rich, sequential memories. We suggest that episodic memories could benefit from moving beyond the idea of keys and values as single vectors. **It can be useful to store more general structures—such as a time-sequence of states—as a single "value" in memory.**

One benefit of episodic memory is that memory replay can support learning [30, 38], in particular by ameliorating catastrophic interference. It would therefore be interesting to explore whether HCAM's memory could be used for training. For example, could memory summaries be used to locate similar or contrasting memories that should be interleaved together [39] to improve continual learning?

**Transformer memories**  Many recent works have attempted to improve the computational efficiency of transformer attention over long sequences [e.g. 28, 58]. However, these approaches often perform poorly at even moderately long tasks [54]. Similarly, we found that on tasks like Ballet or One-Shot StreetLearn, Transformer-XL can perform suboptimally even when the entire task fits within a single, relatively short attention window. However, there could potentially be complementary benefits to combining approaches to obtain even more effective attention with hierarchical memory, which should be investigated in future work. In particular, some recent work has proposed a simple inductive bias which allows Transformers to benefit from evaluation lengths much longer than they were trained on [46]—while this strategy would likely not help with recalling specific instances in detail, it might be complementary to an approach like ours.

Other works have used a hierarchy of transformers with different timescales for supervised tasks [34, 64, 61, 32], for example encoding sentences with one transformer, and then using these embeddings as higher-level tokens. This approach improves performance on tasks like document summarization, thus supporting the value of hierarchy. Luong et al. [36] also showed benefits of both local and global attention computation in LSTM-based models. However, we are not aware of prior transformers in which the coarse computations are used to select chunks for more detailed computation, as in HCAM (although recent work has demonstrated top-$k$ patch selection in CNNs [8]); nor are we aware of prior works that have demonstrated their approaches in RL, or beyond a single task domain.

**Memory and adaptation**  One major benefit of memory is that a model can flexibly use its memories in a goal- or context-dependent way in order to adapt to new tasks [55, 52, 30]. Adult humans use our recall of rich, contextual memories in order to generalize effectively [52, 43]. While our tasks required some forms of goal-dependent memory use—e.g. combining old paths to plan new ones—it would be interesting to evaluate more drastic forms of adaptation. Recent work shows that transforming prior task representations allows zero-shot adaptation to substantially altered tasks [31]. HCAM could potentially learn to transform and combine memories of prior tasks in order to adapt to radically different settings. Exploring these possibilities offers an exciting future direction.

**Limitations & future directions**  While we see our contribution as a step towards more effective mental time-travel for agent memory, many aspects could be further improved. Our implementation requires the ability to keep each previous step in (hardware) memory, even if the chunk containing that step is irrelevant. This approach would be challenging to scale to the entire lifetime of episodic memory that humans retain. Nonetheless, storing the past is feasible up to tens of thousands of steps on current accelerators. This could be extended further by using efficient $k$NN implementations, which have recently been used to perform $k$NN lookup over an entire language dataset [26].

The challenge of scaling to longer-term memory would be helped by deciding what to store, as in some episodic memory models [15], soft forgetting of rarely retrieved memories, and similar mechanisms. However, such mechanisms are simultaneously limiting, in that the model may be unable to recall knowledge that was not obviously useful at the time of encoding. Over longer timescales, HCAM might also benefit from including more layers of hierarchy—grouping memory chunks into higher-order chunks recursively to achieve logarithmic complexity for recall. Even over moderate timescales, more intelligent segmentation of memory into chunks would potentially be beneficial—human encoding and recall depends upon complex strategies for event segmentation [63, 51, 6, 35, 7]. HCAM might also benefit from improved chunk summaries; mean-pooling over the chunk is a relatively naïve approach to summarization, so learning a compression mechanism might be beneficial [47]. These possibilities provide exciting directions for future work.

**Conclusions**  We have proposed a Hierarchical Chunk Attention Memory for RL agents. This architecture allows agents to recall their most relevant memories in detail, and to reason over those memories to achieve new goals. This approach outperforms (or matches the optimal performance of) a wide variety of baselines, across a wide variety of task domains. It allows agents to remember where objects were hidden, and to efficiently learn to navigate in a new neighborhood by planning from memory. It allows agents to recall words they have learned despite distracting intervening tasks, and even across episodes. These abilities to learn, adapt, and maintain new knowledge are critical to intelligent behavior, especially as the field progresses towards more complex environments. We

hope that our work will motivate further exploration of hierarchically structured memories, in RL and beyond. Hierarchical memory may have many benefits for learning, reasoning, adaptation, and intermediate- or long-term recall.

## Acknowledgments and Disclosure of Funding

We would like to acknowledge Adam Santoro, Emilio Parisotto, Jay McClelland, David Raposo, Wilka Carvalho, Tim Scholtes, Tamara von Glehn, Sébastien Racanière, and the anonymous reviewers for helpful comments and suggestions.

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
