# A  Methods

## A.1  Open source attention module

We have open-sourced a Jax/Haiku implementation of our Hierarchical Attention Module, which can be found at: `https://github.com/deepmind/deepmind-research/tree/master/hierarchical_transformer_memory/hierarchical_attention`

We have also released our Ballet and Rapid Word Learning environments, and include links to those and the released versions of other environments we used below.

## A.2  Agent architecture and training

Table 1: Hyperparameters used in main experiments. Where only one value is listed across multiple columns, it applies to all. Where hyperparameters were swept, parameters used for HCAM and TrXL are reported separately.

| | Ballet | Objects | Words | Image | Streets | Associative |
|---|---|---|---|---|---|---|
| All activation fns | ReLU | | | | | |
| State dimension | 512 | | | | | |
| Memory dimension | 512 | | | | | |
| Memory layers | 4 | | | | | |
| Memory num. heads | 8 | | | | | |
| HCAM chunk size | 32 | 16 | 16 | 16 | 4 | 2 |
| HCAM chunk overlap | | - | | | 1 | - |
| HCAM top-$k$ | 8 | 8 | 16 | 16 | 32 | 32 |
| HCAM total num. chunks | always greater than max episode length // chunk size | | | | | |
| Local Attention Window | 64 | 128 | 128 | 128 | 64 | NA |
| TrXL extra length | 256 | 512 | 256 | NA | 200 (full) | NA |
| Visual encoder | CNN | ResNet | | | | |
| Vis. enc. channels | (16, 32, 32) | | | | | |
| Vis. enc. filt. size | (9, 3, 3) | (3, 3, 3) | | | | |
| Vis. enc. filt. stride | (9, 1, 1) | (2, 2, 2) | | | | |
| Vis. enc. num. blocks | NA | (2, 2, 2) | | | | |
| Language encoder | 1-layer LSTM | | NA | | | |
| Lang. enc. dimension | 256 | | NA | | | |
| Word embed. dimension | 32 | | NA | | | |
| Policy & value nets | MLP with 1 hidden layer with 512 units. | | | | | |
| Reconstruction decoders | Architectural transposes of the encoders, with independent weights. | | | | | |
| Recon. loss weight (HCAM) | 1. | | 0.3 | 1. | 0.3 | NA |
| Recon. loss weight TrXL | 1. | | | NA | 1. | NA |
| $V$-trace loss weight (HCAM) | 0.1 | | 0.3 | 0.1 | 1. | NA |
| $V$-trace loss weight (TrXL) | 0.3 | 0.1 | | NA | 1. | NA |
| $V$-trace baseline weight (HCAM) | 1. | | 0.3 | 1. | 0.3 | NA |
| $V$-trace baseline weight (TrXL) | 1. | 0.3 | 1. | NA | 1. | NA |
| Entropy weight | $1 \cdot 10^{-3}$ | $1 \cdot 10^{-4}$ | | | | NA |
| Batch size | 32 | | | | | |
| Training unroll length | 64 | 128 | 128 | 128 | 64 | NA |
| Optimizer | Adam [27] | | | | | |
| LR | $2 \cdot 10^{-4}$ | | | | $4 \cdot 10^{-4}$ | $1 \cdot 10^{-4}$ |

Table 2: Hyperparameter sweeps used in main experiments. The One-Shot StreetLearn and Paired Associative Inference tasks used different chunk size sweeps due to the different task lengths and demands. The Paired Associative Inference tasks do not use RL losses and so did not use the corresponding loss sweeps.

| Reconstruct. loss weight | $\{0.1, 0.3, 1.\}$ |
|---|---|
| $V$-trace loss weight | $\{0.1, 0.3, 1.\}$ |
| $V$-trace baseline weight | $\{0.1, 0.3, 1.\}$ |
| HCAM chunk size | $\{16, 32, 64\}$, except Streets=(2, 4, 8, 16) and Associative=None |
| LR | None, except Streets and Associative=$\{1, 2, 4\}\cdot10^{-4}$ |

In Table 1 we show the hyperparameters used for all experiments, and in Table 2 we show the hyperparameters sweeps used, although we generally used a subset of the full sweep. We swept each hyperparameter with a single seed per condition, and then reran the best parameter settings for each condition with more seeds to get a more robust estimate of the performance of each approach.

In most cases the hyperparameters that were not swept were taken from other sources without tuning for our architecture. In particular, the first tasks we considered were the rapid word learning tasks, and many parameters were taken directly from the hyperparameters of the original paper on which the tasks were based [19]. These hyperparameters were therefore tuned directly by prior researchers for other models, but we found them to work well for our memory as well. The visual encoder, language encoder, unsupervised reconstruction loss etc. were copied from those described in the prior work.

Since we ran all other experiments after the initial word learning experiments, we used many of these same hyperparameters on other experiments, such as the visual and language encoder architecture across all experiments. However, some hyperparameters do differ across tasks due to specific task features. For example, the visual encoder for the ballet tasks is set to have a filter size of 9 because this is the resolution of each square in the grid, and the entropy cost for the ballet tasks was chosen from our prior work [18] which used a similar grid world action space. These decisions were shared across all architectures, so should not favor our model over the baselines.

**Self-supervised reconstruction loss**    We used the same reconstruction loss as Hill et al. [19], namely reconstructing the language with a softmax cross-entropy loss, and reconstructing the image pixels (normalized to range [0, 1] on each color channel) with a sigmoid cross-entropy loss. The image reconstruction loss was averaged across all pixels and channels, while the language reconstruction loss was summed across the sequence.

### A.2.1   Bug fixed between original and revised versions of this paper

Shortly before the camera-ready deadline for NeurIPS, we discovered a bug in the configuration of the HCAM in the Ballet, Words, and Passive Visual Match domains: the local attention window was much longer than intended. Fixing this bug did not substantially alter results in the Ballet or Passive Visual Match tasks, but did change our results somewhat in the Rapid Word Learning tasks. The qualitative patterns of extrapolation and generalization to multiple episodes remain the same, but generalization of HCAM is somewhat worse, although still much better than the baseline models. This does not substantially affect the conclusions of the paper. We have revised the Rapid Word Learning plots in the main text to reflect these updated results, and included evaluation on the original levels in Fig. 7. However, note that our supplemental analyses in this domain were carried out with the longer attention window.

### A.3   Plotting methods

In all plots, each curve is an average across multiple runs. The $x$-axis is always the number of agent steps (actions taken/frames seen) during training. The number of learner updates is generally $2000 - 4000\times$ smaller with a batch size of 32 trajectories per update, and unroll lengths of 64-128. The dark regions around the curve show $\pm$SD across runs, the light regions show the total range. The plots are smoothed by interpolation with a triangular window, with width and sampling frequency chosen to present results clearly depending on the speed and variability of learning the different tasks. All figures were made with seaborn [59] and matplotlib [22].

### A.4 Compute resources

All experiments were run using Google TPU v2, v3, and v4 devices. Each run lasted between a few hours and a few days depending on the experiment. We ran actors/evaluators on CPUs. We estimate the total time needed to reproduce all experiments (including baselines and experiments in appendices) to be around 1000 TPU-hours + 300000 CPU hours.

## B Tasks

We have uploaded selected video recordings of the HCAM-based agent performing our main tasks at `https://www.youtube.com/playlist?list=PLE5lx5-YU_Hr8Q9IgTAfisJ6XCy3Jhh6F`

### B.1 Open source or released tasks

We have open-sourced our Ballet environment at: `https://github.com/deepmind/deepmind-research/tree/master/hierarchical_transformer_memory/hierarchical_attention`

We have released our rapid-word-learning tasks in the repository for the paper they were based upon `https://github.com/deepmind/dm_fast_mapping`

The environments from other papers that we used also have corresponding releases:

1. Passive Visual Match: `https://github.com/deepmind/deepmind-research/tree/master/tvt`
2. Paired Associative Inference: `https://github.com/deepmind/deepmind-research/tree/master/memo`
3. One-Shot StreetLearn `https://github.com/deepmind/deepmind-research/tree/master/rapid_task_solving`

### B.2 Ballet

The tasks took place in a $9 \times 9$ tile room with an extra 1 tile wall surrounding on all sides, for a total of $11 \times 11$ tiles. This was upsampled at a resolution of 9 pixels per tile to form a $99 \times 99$ image as input to the agent. The agent was placed in the center of the room, and the dancers were placed randomly in 8 possible locations around it. The dancers always had distinct colors and shapes, selected from 15 shapes and 19 colors. These features merely served to distinguish the dancers. The agent always appeared as a white square. The agent received egocentric inputs (that is, its visual input was centered on its location), as this can improve generalization [18].

In Listing 1 we show the dance sequences used for the ballet tasks. All dances are 16 steps long. We trained all agents with levels uniformly sampled to have 16 or 48 steps of delay between dances, and 2, 4, or 8 dances. The number of dancers in the room corresponded to the number of dances, such that if there were only 2 dances, there were only 2 dancers, while if there were 8 dances there were 8 dancers. This is why chance-level performance is 50% with 2 dances, but 12.5% with 8. The agent was given a reward of 1 for a correct choice, and 0 for an incorrect choice.

### B.3 Object permanence

The tasks took place within a 3D environment created with Unity. The agent received a visual observation of $96 \times 72 \times 3$ pixel RGB images, and a language observation that was tokenized at the word-level. The agent was initially placed in a fixed position near one wall of the room facing toward the center, and the boxes were randomly placed within the agent's field of view. When each object appeared, it jumped out of its box three times in succession. If there was a delay period, it began after the object returned to its box for the third time. The delay periods we used for the varying length training were 0, 10, 20, and 30 seconds. After the last presentation and delay phase, the lids of the boxes closed.

After the lids of the boxes closed, the agent was allowed to move and look around, and was given the instruction "look backward." The agent was rewarded 0.3 for looking backwards (far enough that the

```
{
"circle_cw": [0, 2, 4, 4, 6, 6, 0, 0, 2, 2, 4, 4, 6, 6, 0, 2],
"circle_ccw": [0, 6, 4, 4, 2, 2, 0, 0, 6, 6, 4, 4, 2, 2, 0, 6],
"up_down": [0, 4, 4, 0, 0, 4, 4, 0, 0, 4, 4, 0, 0, 4, 4, 0],
"left_right": [2, 6, 6, 2, 2, 6, 6, 2, 2, 6, 6, 2, 2, 6, 6, 2],
"diagonal_uldr": [7, 3, 3, 7, 7, 3, 3, 7, 7, 3, 3, 7, 7, 3, 3, 7],
"diagonal_urdl": [1, 5, 5, 1, 1, 5, 5, 1, 1, 5, 5, 1, 1, 5, 5, 1],
"plus_cw": [0, 4, 2, 6, 4, 0, 6, 2, 0, 4, 2, 6, 4, 0, 6, 2],
"plus_ccw": [0, 4, 6, 2, 4, 0, 2, 6, 0, 4, 6, 2, 4, 0, 2, 6],
"times_cw": [1, 5, 3, 7, 5, 1, 7, 3, 1, 5, 3, 7, 5, 1, 7, 3],
"times_ccw": [7, 3, 5, 1, 3, 7, 1, 5, 7, 3, 5, 1, 3, 7, 1, 5],
"zee": [1, 6, 6, 2, 2, 5, 1, 5, 5, 2, 2, 6, 6, 1, 5, 1],
"chevron_down": [7, 4, 3, 1, 0, 5, 1, 5, 1, 4, 5, 7, 0, 3, 7, 3],
"chevron_up": [3, 0, 7, 5, 4, 1, 5, 1, 5, 0, 1, 3, 4, 7, 3, 7],
}
```

Listing 1: Dances used in the ballet task. 0-7 refer to directions of movement, clockwise from 0 = up.

boxes were out of view), and looking backward allowed it to advance to the choice phase of the task. In the choice phase, the agent was told "go to the box containing the [duck]" and was rewarded 1 for making the correct choice, and 0 for an incorrect choice.

### B.4 Rapid word learning with distractors

We used the tasks created by Hill et al. [19] with the following modifications. First, we removed three of the possible objects (trains, robots, and rockets) to be used in the distractor task. We then added 0-20 distractor phases between the word binding and test phases. In each distractor phase, the agent and the three distractor objects were randomly placed in the room, and the agent was asked to lift one of them, e.g. "lift the rocket." The agent received a reward of 0.1 for successfully lifting the right object, and was allowed to progress to the next distractor task. If the agent lifted the wrong object, it was neither rewarded nor allowed to progress until it had lifted the correct object or until 20 seconds had passed. All agents rapidly learned to solve these distractor tasks. A fixed time limit of 450 seconds was used across all episodes, after which the episode terminated with reward 0 regardless of what phase the agent was in.

For the multi-episode evaluation tasks, we simply combined the number of episodes we wished to test across into a single "super-episode." For the final test phase, where we tested earlier words, the distractor objects were always taken from the same learning phase as the target object (to ensure that the agent remembered the exact name-object pairings, rather than simply which name appeared with which group of objects). We set a time limit of 450 seconds to complete *all* the sub-episodes. Agents with HCAM and TrXL memories were able to consistently complete the super-episodes within this time limit—even though TrXL could not choose the correct objects, it was consistently reaching the end and choosing *some* object for a chance at the final reward. However, the LSTM-based agents often timed out on these multi-episode evaluations.

### B.5 Comparisons to other papers

The Passive Visual Match and Paired Associative Inference tasks were used unmodified. The StreetLearn [40] images and maps we used for the One-Shot StreetLearn were a more recent version than those used by Ritter et al. [49]. Because the task difficulty is fixed through the sampling of neighborhoods from the larger city graph, this should not substantially alter the difficulty of the tasks. We received permission from an author on each paper to use their tasks.

All three tasks from other papers have been released under Apache licenses, and the open source code can be found at:

- Passive visual match: `https://github.com/deepmind/deepmind-research/tree/master/tvt/dmlab`

**The PAI task** In order to apply HCAM to the supervised PAI task, we took the following steps. We embedded all the input memories and probes using a single shared embedding layer. The structure of the memories for the PAI task matches the structure of HCAM's contents, where each pair of associated images corresponds to a single chunk in memory (of length 2). We therefore created a HCAM-style memory containing these embedded contents, and keyed by their summaries (averages across each chunk). We then provided the embedded query as input to the multi-layer HCAM model, but used the same set of embedded task memories at every layer. After 4 HCAM layers, we averaged-pooled across the sequence of resulting embeddings, and then performed a linear projection to produce a final output embedding. We then compared this output embedding to the embeddings of the two possible choices using dot products. These dot products were used as logits in a softmax to choose the answer, and the model was trained using a cross-entropy loss. We did not use a self-supervised reconstruction loss for this setting.

## C  Detailed results

In Table 3 we show the mean performance and standard deviation across runs from our main experiments.

Table 3: Numerical results from main experiments/figures—mean $\pm$ standard deviation across 3 runs per condition. Results are average performance (% correct) across evaluations during the last 1% of training, except for the One-Shot StreetLearn tasks, where they are average reward during the last 1% of training. (Note that on some levels LSTMs were not consistently completing the task before the episode time limit. Incomplete episodes are scored as 0.)

| Experiment | Level | Fig. | HCAM | TrXL | LSTM |
|---|---|---|---|---|---|
| Ballet | 2 dances, delay 16 | 3a | $99.8 \pm 0.3$ | $96.9 \pm 1.4$ | $97.4 \pm 1.8$ |
| | 8 dances, delay 16 | 3b | $98.1 \pm 3.3$ | $65.7 \pm 13.7$ | $25.2 \pm 2.7$ |
| | 8 dances, delay 48 | 3c | $97.2 \pm 2.5$ | $49.2 \pm 13.0$ | $29.7 \pm 9.4$ |
| Objects | No delay, varying train | 4b | $96.7 \pm 0.9$ | $82.2 \pm 28.7$ | $33.2 \pm 6.6$ |
| | Long delay, varying train | 4c | $91.7 \pm 8.3$ | $46.1 \pm 20.0$ | $34.8 \pm 5.5$ |
| | Long delay, long-only train | 4d | $82.9 \pm 16.4$ | $31.1 \pm 0.6$ | - |
| Words | 10 distractors | 5d | $93.0 \pm 4.0$ | $76.7 \pm 12.5$ | $21.0 \pm 18.3$ |
| | 4 episodes, 0 distractor each | 5e | $82.8 \pm 6.4$ | $49.3 \pm 16.5$ | $19.8 \pm 17.4$ |
| | 2 episodes, 5 distractors each | 5f | $71.1 \pm 8.0$ | $48.7 \pm 13.2$ | $22.9 \pm 20.1$ |
| Image | | 6a | $97.0 \pm 2.8$ | - | - |
| Associative | | 6b | $97.5 \pm 0.9$ | - | - |
| Streets | | 6c | $26.8 \pm 0.44$ | $19.9 \pm 0.65$ | - |

## D  Supplemental experiments

In this section we present some supplemental experiments and analyses. However, we make several notes here. First, these supplemental analyses were mostly run with a longer local attention window than used in the main text, see App. A.2.1, which could potentially affect results, particularly in the rapid word learning domain. Second, we use the original acronym HTM instead of HCAM in most of these plots, because we revised it only after a reviewer pointed out a name clash.

### D.1  Fast-binding performance on harder hold-out tasks

In our original version of this paper, we presented generalization results on a harder set of evaluation tasks. Unfortunately, the high generalization performance on these results seemed to be at least in part due to a bug causing our HCAM memory to have a large local attention window (see above). We therefore changed the main text figures to show performance on slightly easier task variations.

However, in Fig. 7 we show performance on the original evaluation tasks. HCAM still achieves off-chance performance in 2 out of 3 cases, with fairly decent performance in one case, and performance continues to improve as training goes on.

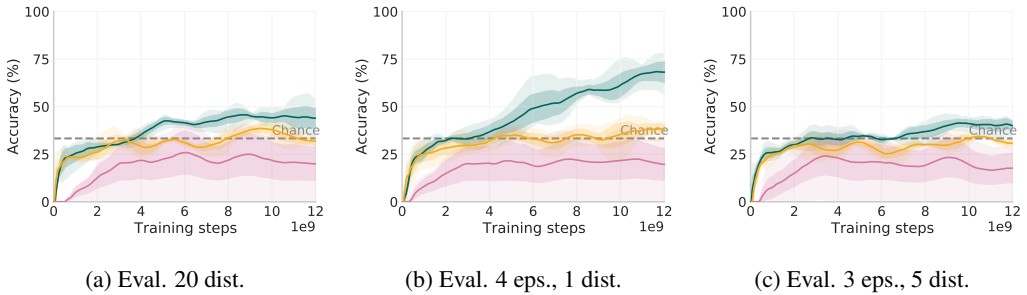

(a) Eval. 20 dist.                    (b) Eval. 4 eps., 1 dist.                    (c) Eval. 3 eps., 5 dist.

Figure 7: Evaluating HCAM on the harder generalization tasks we considered in the original version of this paper, after longer training (note horizontal axis). HCAM achieves off-chance performance, and continues to improve as training goes on. (3 seeds per condition.)

## D.2 Ballet generalization

In the main text Ballet experiments (Fig. 3), we compared differences only in training performance. In Fig. 8, we show that HCAM is also able to generalize well from training on 2, 4, or 6 dances, to evaluation on 8 dances with either short or long delays.

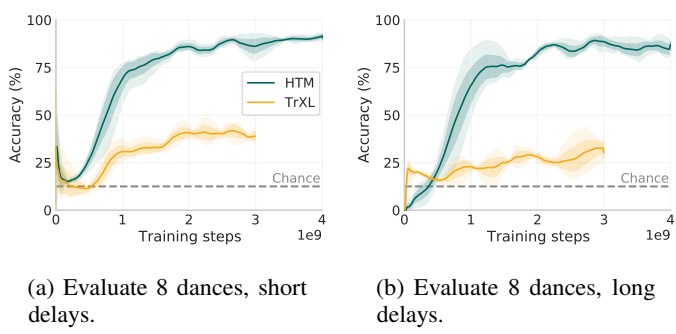

(a) Evaluate 8 dances, short delays.                    (b) Evaluate 8 dances, long delays.

Figure 8: HCAM (labeled as HTM) trained on 2, 4, or 6 dance ballets generalizes well to 8 dance ballets, while TrXL does not. Results are from two seeds in each condition.

## D.3 Analyzing memory attention in the rapid word learning tasks

In this section, we show that the HCAM agent's extrapolation to cross-episode evaluation (Fig. 5b) in the rapid word learning tasks is supported by selective patterns of memory access. In summary, we show that when the agent is asked to recall a word from many episodes before, all layers of its memory exhibit significantly higher attention to phase when it learned that word, compared to its attention patterns when tested on a more recent word.

Specifically, we considered the case of evaluation across 4 episodes, with 1 distractor each. To lay out the "super-episode" structure of this setting very explicitly, it proceeds through 4 episodes, each of which has three distinct phases. In other words, the episodes proceed as: learn 1, distract 1, test 1, learn 2, distract 2, test 2, learn 3, distract 3, test 3, learn 4, distract 4, test 4. Tests 1-3 evaluate memory for a word learned in their respective learn phases 1-3, but test 4 tests surprises the agent by asking about a word learned in learning phase 1 (Fig. 5b). As shown in the main text (Fig. 5e), the HCAM-based agent achieves above 90% performance on test 4, despite never being evaluated on words from earlier episodes during training.

To investigate the attention patterns underlying this result, we analyzed what chunks of memory the agent was attending to in test phase 4 (Fig. 9). In particular, we ran the agent on 100 of these episodes,

and saved its attention weights for each phase of the experiment. In 93 of these episodes, the agent chose correctly in test 4. Within those 93 episodes, we then evaluated the agent's attention to the first memory chunk of learn 1, the learning phase of the first episode (which generally contained most, if not all, of that learn phase). We compare the attention weight on this chunk when the agent is tested on one of these words in test phase 4 to a within-episode control: the relative weight when the agent is tested on a word from learn 3 during test phase 3. Is the agent attending more to its memory of learn phase 1 in test 4, when that memory is relevant, compared to test 3, when it is irrelevant? In fact, we find that across all four layers of the agent's memory, the agent is attending more strongly to the memory when it is relevant than when it is not. That is, the agent is distributing its attention intelligently, in a query-dependent way. It is attending most strongly to memories of the learn 1 phase when it is asked to recall a word from it, compared to a within-super-episode control where it is asked to recall a word from another phase (learn 3).

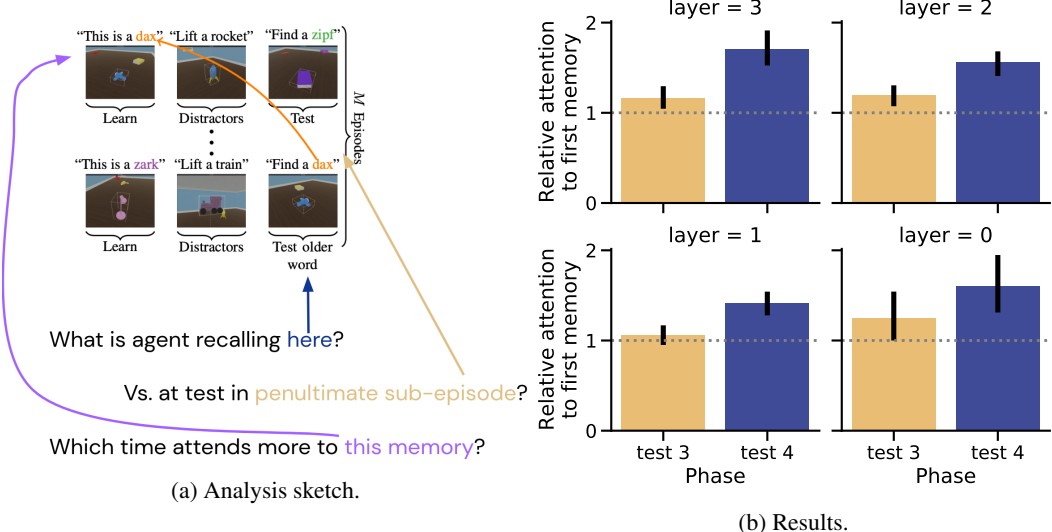

(a) Analysis sketch.

(b) Results.

Figure 9: The HCAM-based agent selectively attends to relevant memories in the rapid-word-learning generalization tasks. (a) We analyze the relative weight of attention to the first memory from the first learning phase, when the agent is asked to recall a word form the first phase in test 4 vs. when it is asked to recall a word from a later phase in test 3. (b) Across all 4 memory layers, the agent attends more strongly to its memory of this first learning phase when that memory is relevant—in test 4—compared to when that memory is irrelevant—in test 3. (This plot shows relative attention weights—that is, attention weights divided by average attention weight, so that if the agent were attending uniformly to all memories, their relative attention weights would be 1, indicated by the dotted line. This plot shows averages and 95%-CIs across the 93 episodes where the agent made a correct choice in test 4, out of 100 total super-episodes run.)

The results above involve several levels of aggregation: averaging within phases, and across many super-episodes. To give a flavor for the complexity of the full patterns of attention, in Fig. 10 we show the average attention weights for every layer across all the stored memories, in the final four phases of two super-episodes.

## D.4    Varying chunk sizes

In Fig. 11 we show that the performance of the HCAM model is robust to varying chunk sizes in the ballet task; therefore its advantage in this task is not due to having additional information about the correct segmentation of the episodes. Furthermore, HCAM performs well even when its chunk size is 12, and so the total number of timepoints it can attend to at each layer is smaller than the number that the TrXL can attend to at each layer. Thus its advantage in these tasks is not due to attending to more of the episode, but rather to attending more effectively.

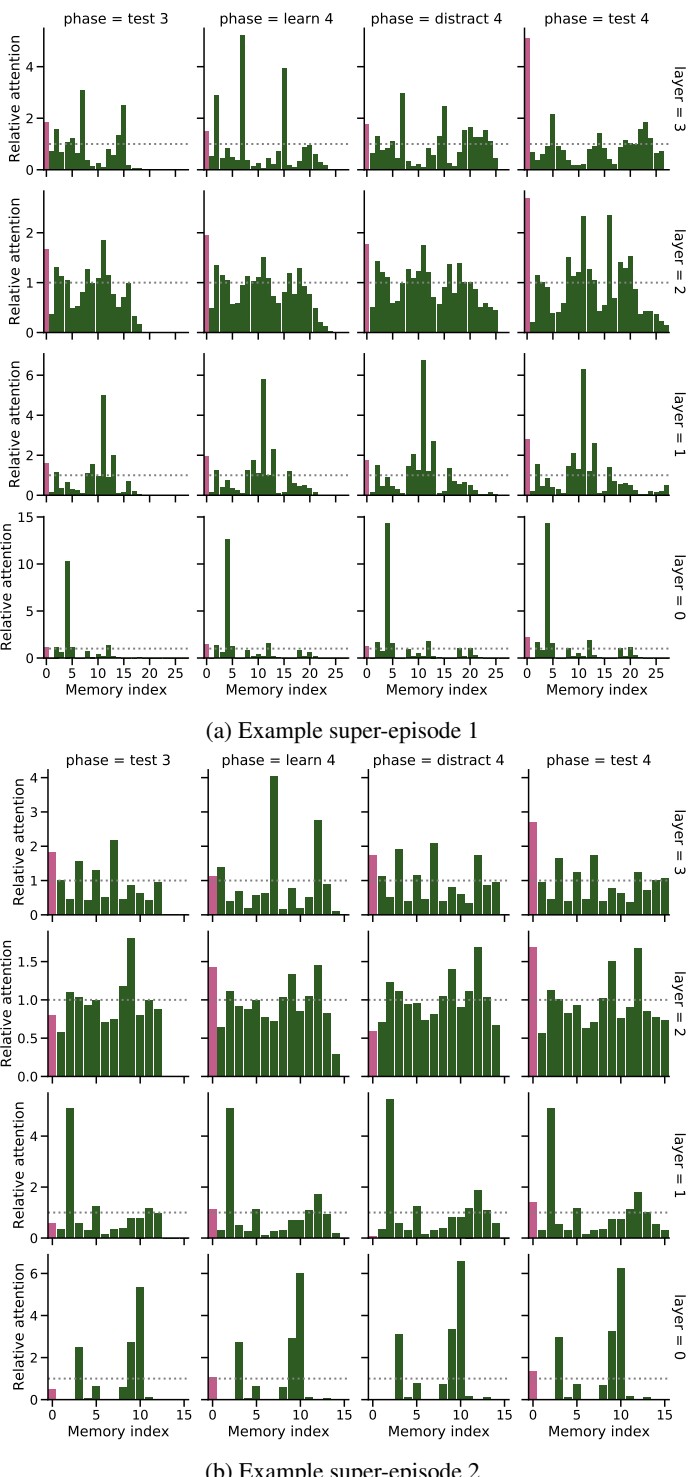

(a) Example super-episode 1

(b) Example super-episode 2

Figure 10: Patterns of attention within four phases of two example (randomly chosen) super-episodes on the rapid-word-learning generalization tasks. The higher layers of the network show clear shifts in attention patterns between the different phases, although with some consistent biases within each super-episode, especailly at the lower layers. The pink bar shows the weight on the first chunk from learn phase 1—the analyis shown in Fig. 9 corresponds to comparing the pink bars in the first and last column, aggregated across many more episodes. (This plot shows relative attention weights— that is, attention weights divided by average attention weight, so that chance level relative attention would be 1, indicated by the dotted line. Note that these were computed *before* the top-$k$ operation on the attention weights, which is why more than 16 weights are active. The two super-episodes had different lengths, which is why more memories were stored in the first.)

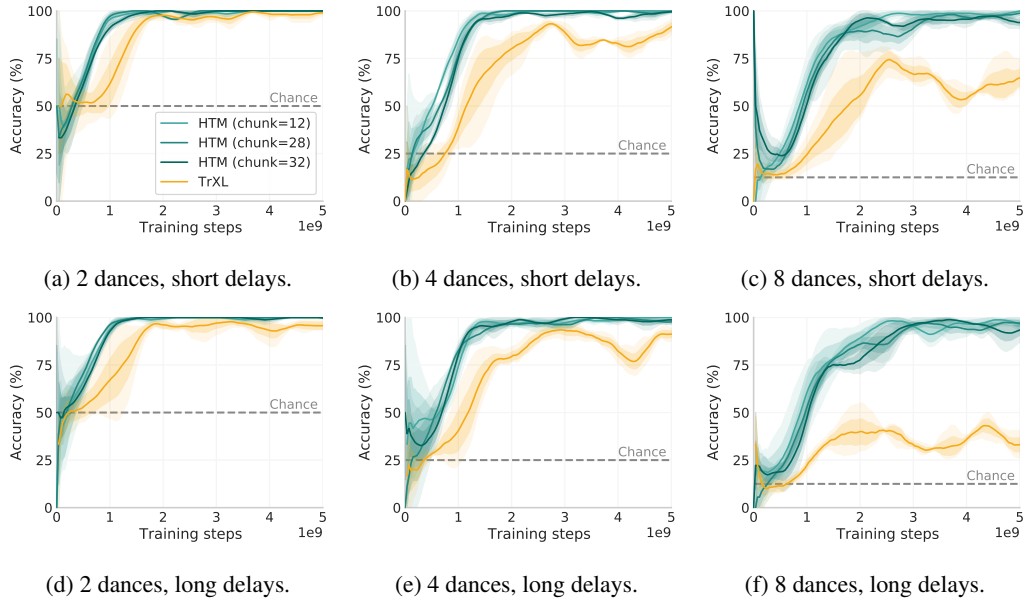

(a) 2 dances, short delays.  (b) 4 dances, short delays.  (c) 8 dances, short delays.

(d) 2 dances, long delays.  (e) 4 dances, long delays.  (f) 8 dances, long delays.

Figure 11: Comparison of HCAM (labeled as HTM) with different chunk sizes to TrXL across the different ballet levels. The performance of the HCAM model is robust to varying chunk size, indicating that HCAM does not need a task-relevant segmentation to perform well. The results reported in the main text use chunk size 32; panels a, c, and f correspond to main text Fig. 3. These comparisons were run before a minor bug was fixed in HCAM memory writing. Results are from three seeds in each condition.

### D.5  Varying $k$ for memory selection

In Fig. 12 we show that HCAM is robust to varying the number $k$ of memory chunks selected in the top-$k$ step of the hierarchical attention at each layer. Specifically, while the main text experiments used $k = 16$, we show that HCAM is able to perform the ballet and object permanence tasks well even with $k = 4$, and can perform the shorter tasks even with $k = 2$ or even $k = 1$. While it initially surprised us that hard memory selection with $k = 1$ did not harm the optimization process, it resonates with recent results from the Switch Transformer [12], which found that hard selection of a single expert was effective in a mixture-of-experts style model.

### D.6  The importance of self-supervised learning

In Fig. D.6, we show that the self-supervised loss (image + language reconstruction at the agent output) that we used as an auxiliary loss during training is necessary for our model to achieve good performance on the ballet and fast-binding tasks. This is presumably because this loss forces the model to encode the input in detail, and therefore that information is in-principle retrievable from the state representations stored in the agent memory.

### D.7  Memory layer gating

In Fig. 14, we show that HCAM's performance is not enhanced by the gating mechanism proposed by Parisotto et al. [44], and in fact HCAM actually learns slightly more slowly when its layers are gated. This is potentially because HCAM already has some notion of gating in its selection of relevant chunks, and additional gating therefore only interferes with the optimization process. Thus, we did not use gating for HCAM in our main experiments. Furthermore, we found that gating was not necessary to train the TrXL memory on our tasks, although we used it in our main experiments to match Parisotto et al. [44]. However, HCAM with or without gating outperforms TrXL with or without gating.

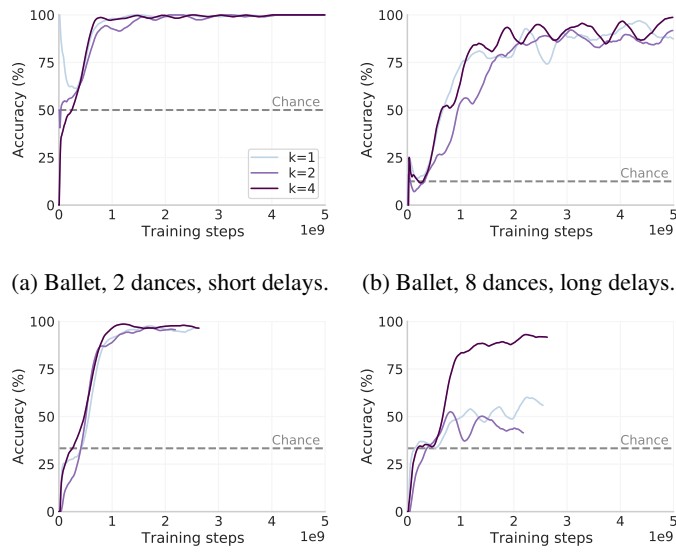

(a) Ballet, 2 dances, short delays.     (b) Ballet, 8 dances, long delays.

(c) Object permanence, no delays.   (d) Object permanence, 30s delays.

Figure 12: Varying the number $k$ of memory chunks selected in the top-$k$ step of hierarchical attention. Performance is relatively robust to $k$ smaller than used in the main experiments ($k = 16$), in fact shorter tasks can be learned even with $k = 1$, while longer tasks require $k \geq 4$. (a-b) On the ballet tasks, HCAM can learn fairly well even with $k = 1$, both for the shorter and longer tasks. (c-d) On the object permanence tasks, HCAM can learn the shortest tasks well even with $k = 1$, but struggles to learn the longer tasks unless $k \geq 4$. (One seed per condition.)

## D.8    Comparing a TrXL that is $2\times$ wider/deeper than HCAM

Because our HCAM-based agents have an added HCAM attention block in each memory layer compared to our TrXL-based ones, it might seem that they have somewhat more parameters and greater total depth. However, as noted in Section D.7, HCAM does not use the gating layers used by the TrXL memory [44] and because of this HCAM uses about 20% fewer parameters and is in some sense shallower than our TrXL baselines. However, it does have more layers of attention. To ensure that this or other simple factors were not the primary driver of HCAM's advantage, we ran comparisons where we either made the TrXL twice as wide (i.e. each layer had twice as many hidden units, including the attention projections etc.) or twice as deep (i.e. an 8-layer TrXL memory instead of 4-layers as we used for our main experiments). Both of these have substantially more parameters than our HCAM-based models, and the latter is substantially deeper as well. However, we show in Fig. 15 that these much larger TrXL models were also unable to match the performance of HCAM on the rapid word-learning tasks. Thus the advantage of HCAM is not due to parameters or depth alone.

## D.9    Sparsity without hierarchy: a top-$k$ TrXL

One possible explanation of our results would be that sparsity alone is sufficient—perhaps the TrXL is suffering from spreading its attention across too many points in the past, but if it were restricted to only a few points hierarchy would not be necessary. To evaluate this possibility, we created a modified TrXL where we imposed sparsity of attention, by truncating its attention to only the top-$k$ most relevant timepoints. We chose $k = 16$ to match HCAM. We show the results in Fig. D.9. The top-$k$ TrXL performs comparably to a standard TrXL on the ballet tasks (i.e. does not perform as well as HCAM), and fails to learn properly on the rapid word-learning tasks, even if allowed to attend to a larger number of points ($k = 32$). Thus, sparsity without hierarchy does not suffice, and may actually harm learning.

## D.10    Compute efficiency assessed by learner FPS

One goal of HCAM is that sparser attention might be more efficient than full attention. This is especially true when comparing HCAM without gating to the more computationally intense Gated

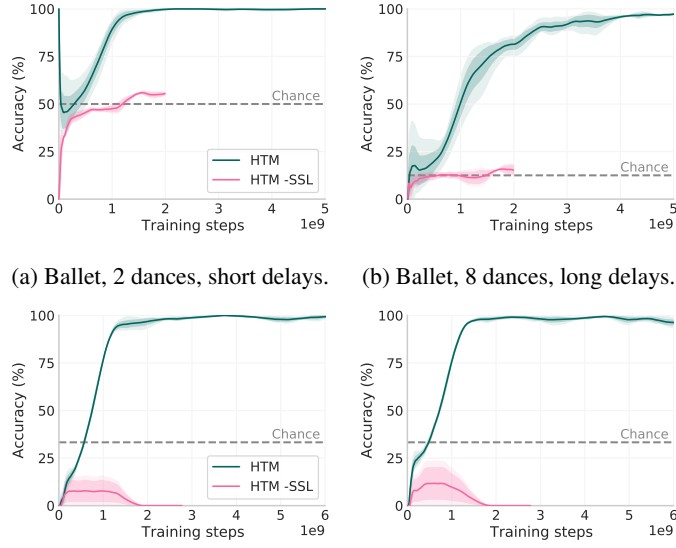

(a) Ballet, 2 dances, short delays.     (b) Ballet, 8 dances, long delays.

(c) Rapid word learning, no distrac- (d) Rapid word learning, 2 distrac-
tors.     tors.

Figure 13: The self-supervised loss (SSL) is necessary for the model to learn appropriate representations. When the SSL is disabled, the model either fails to achieve substantially-above-chance performance, for example in the ballet tasks (a-b), or fails to learn the tasks to even chance level, as in the rapid word learning tasks (c-d). (HTM refers to HCAM, see note above. 3 runs per condition for main results, 2 runs per condition for results without SSL.)

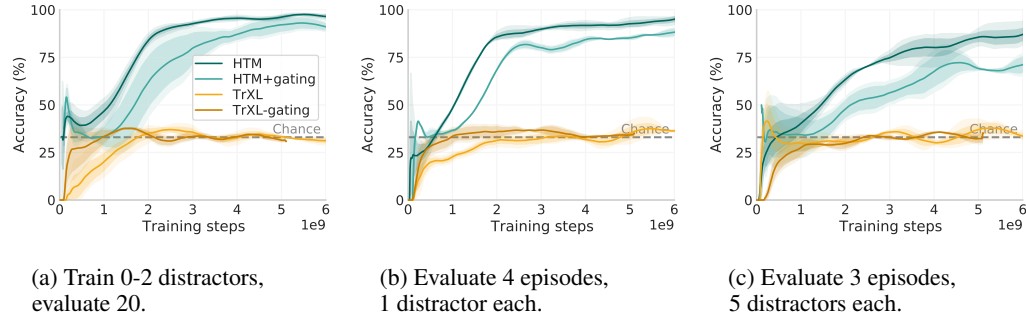

(a) Train 0-2 distractors,     (b) Evaluate 4 episodes,     (c) Evaluate 3 episodes,
evaluate 20.     1 distractor each.     5 distractors each.

Figure 14: HCAM (labeled as HTM) performs better without gating [44] than with gating. On the fast-binding tasks HCAM with gating learns slightly more slowly and generalizes slightly worse than without gating. Gating of memory layers does not appear necessary for TrXL in our tasks, unlike the experiments of Parisotto et al. [44]. However, neither gated nor ungated TrXL are able to extrapolate to the tasks that gated or ungated HCAM does. (3 seeds per condition for main runs, 2 per condition for alternatives.)

TrXL [44]. Correspondingly, we show in Fig. 17 that HCAM generally runs ∼30-40% faster than TrXL.

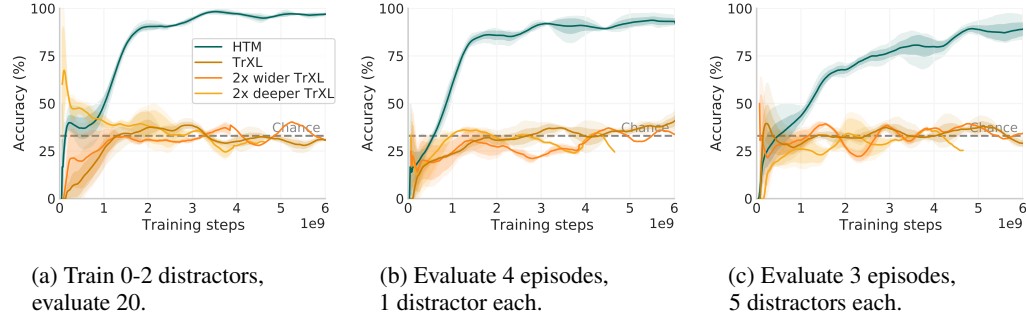

(a) Train 0-2 distractors, evaluate 20.

(b) Evaluate 4 episodes, 1 distractor each.

(c) Evaluate 3 episodes, 5 distractors each.

Figure 15: Comparing agents with TrXL memories that have more parameters than HCAM on the rapid word-learning tasks. Neither a TrXL model that is twice as wide, nor one that is twice as deep are able to perform as well as HCAM. Thus, HCAM's advantage is not due to the added blocks or slightly more parameters than our TrXL baseline. (HTM refers to HCAM, see note above. 3 seeds per main condition, 2 seeds per condition for supplemental.)

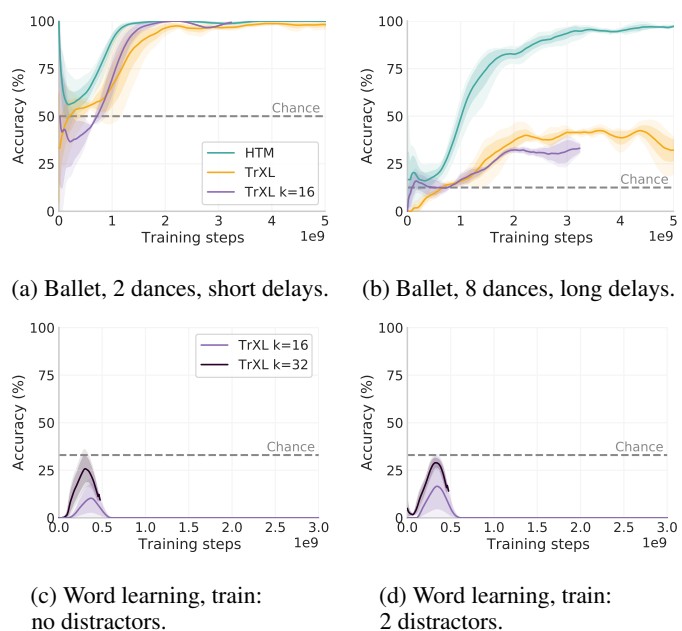

(a) Ballet, 2 dances, short delays.

(b) Ballet, 8 dances, long delays.

(c) Word learning, train: no distractors.

(d) Word learning, train: 2 distractors.

Figure 16: Sparsity alone is not sufficient—a TrXL restricted to attend to only the top-$k$ timepoints performs comparably to a standard TrXL at the ballet tasks (a-b), but collapses and fails to learn in the rapid word learning tasks (c-d), even if given a larger $k$. The advantage of HCAM is not due to sparsity alone. (HTM refers to HCAM, see note above. 3 seeds per main condition, 2 seeds per condition for supplemental.)

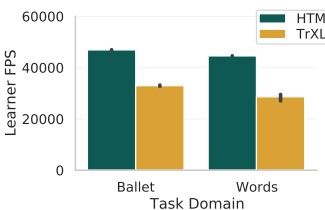

Figure 17: HCAM (labeled as HTM) runs at a higher speed (measured in average learner frames processed per second) than TrXL. Results on TPUv3, error bars show 95%-CI across 3 runs.

### D.11 Results are robust to reviewer-suggested hyperparameter sweeps

Our reviewers evaluated the paper carefully, and expressed concerns that there might be bias in our hyperparameter selection. In particular, one reviewer raised a concern that TrXL might learn tasks like Ballet if given a different learning rate. To address these concerns, we ran a set of follow-up hyperparameter sweeps. We swept the learning rate and entropy weight (which we had not varied previously from the values used in prior work) on both the Ballet and Word tasks. We ran a full product of three learning rates above/below our original values (5e-4, 5e-5, 1e-5) and entropy weights $5\times$ more or less than the original value, with 2 seeds per condition (for a total of 24 hyper $\times$ seed $\times$ memory type combinations per task domain). We emphasize that the same hyperparameters were tested for both models, and that these sweeps centered on hyperparameter settings that were previously untuned (sourced from prior papers), and that this sweep focuses on learning rate, which a reviewer suggested might particularly benefit TrXL. Our results show that HCAM is more robust to variation in these hyperparameters than TrXL, and generally sweeping these parameters does not improve TrXL's performance substantially beyond the results reported in the main text.

**Ballet results:** HCAM substantially outperforms TrXL in this sweep as well. First, HCAM is far more robust to varying the hyperparameters—in every hyperparameter setting, agents with HCAM achieved off chance performance (measured as window-averaged performance >5 percentage points above chance-level) on the hard 8-dance tasks within 1 billion steps. By contrast, 58% of the TrXL jobs did not attain off chance performance on even the easiest task within 1.5 billion steps (when we stopped the training). In addition, 66% of the HCAM jobs achieved above-75% performance on the easiest tasks before *any* of the TrXL jobs achieved above-chance performance on any task. The performance of the best TrXL jobs from this sweep is comparable to the performance at the same point in training from our original experiments: about 40-50% performance on the 8 dance, short delays task, and 25-35% on the 8 dance, long delays task at 1.5 billion steps. HCAM performed much better than TrXL, with 75% of the HCAM jobs outperforming even the best TrXL agents on the hardest task, and the best HCAM jobs comparable to the results in the paper, achieving 80-90% performance on the hardest tasks at 1.5 billion steps. In both 8 dance levels, the advantage of the two HCAM seeds with the best hyperparams over the two TrXL seeds with the best hyperparams is significant by a paired[1] t-test, respectively $t(1) = 21, p = 0.03$ and $t(1) = 101, p = 0.006$. In summary, TrXL's performance at these tasks does not seem to be improved by varying learning rates or entropy weight, and HCAM seems much more robust to variation in these hyperparameters.

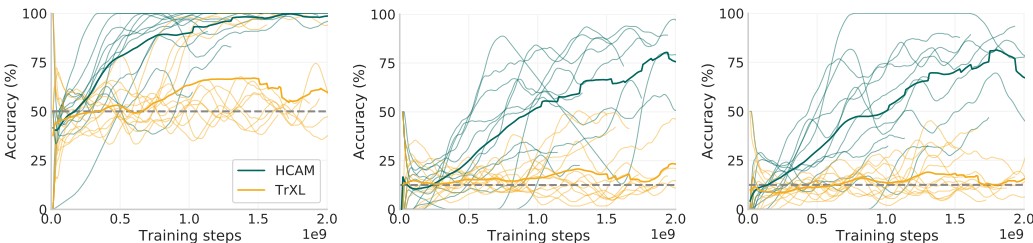

(a) Ballet, 2 dances, short delays.  (b) Ballet, 8 dances, short delays.  (c) Ballet, 8 dances, long delays.

Figure 18: Sweeping hyperparameters (learning rate and entropy weight) in the Ballet tasks: HCAM is substantially more robust to variation in these hyperparameters, and the conclusions of our main text experiments are unaltered. The thick line plots the mean, while the thin lines plot individual sweep values. (The wide variability in early accuracy values should be disregarded—it is due to smoothing artifacts due to sparse data in this region as evaluation jobs are starting.)

**Rapid word learning results:** The results are similar to the above. First, HCAM is more robust to varying hyperparameters: In these more challenging tasks, only 17% of the TrXL jobs achieve high training performance within 5 billion steps, while 50% of the HCAM jobs achieve high performance on the training tasks. HCAM also generalizes better than TrXL. However, unlike our original experiments, one set of these TrXL jobs does achieve somewhat above-chance performance at one of

---

[1] paired reflecting the non-independence of the encoder initialization when the agents are initialized with the same random seed, but results are similar with an unpaired test, respectively $t(2) = 29, p = 0.001; t(2) = 9.5, p = 0.01$

the the evaluation tasks we considered. We performed a replication with three new random seeds in the best hyperparameters from this sweep for each memory (as in the main results), and in this replication the TrXL did not achieve significantly off chance performance. However, HCAM did achieve significantly off-chance performance, (though not as high as the main text results using the hyperpareters tuned in our original sweeps). Thus, HCAM again appears to be both more robust across hyperparameters, and better when comparing best-hyperparameter configurations.

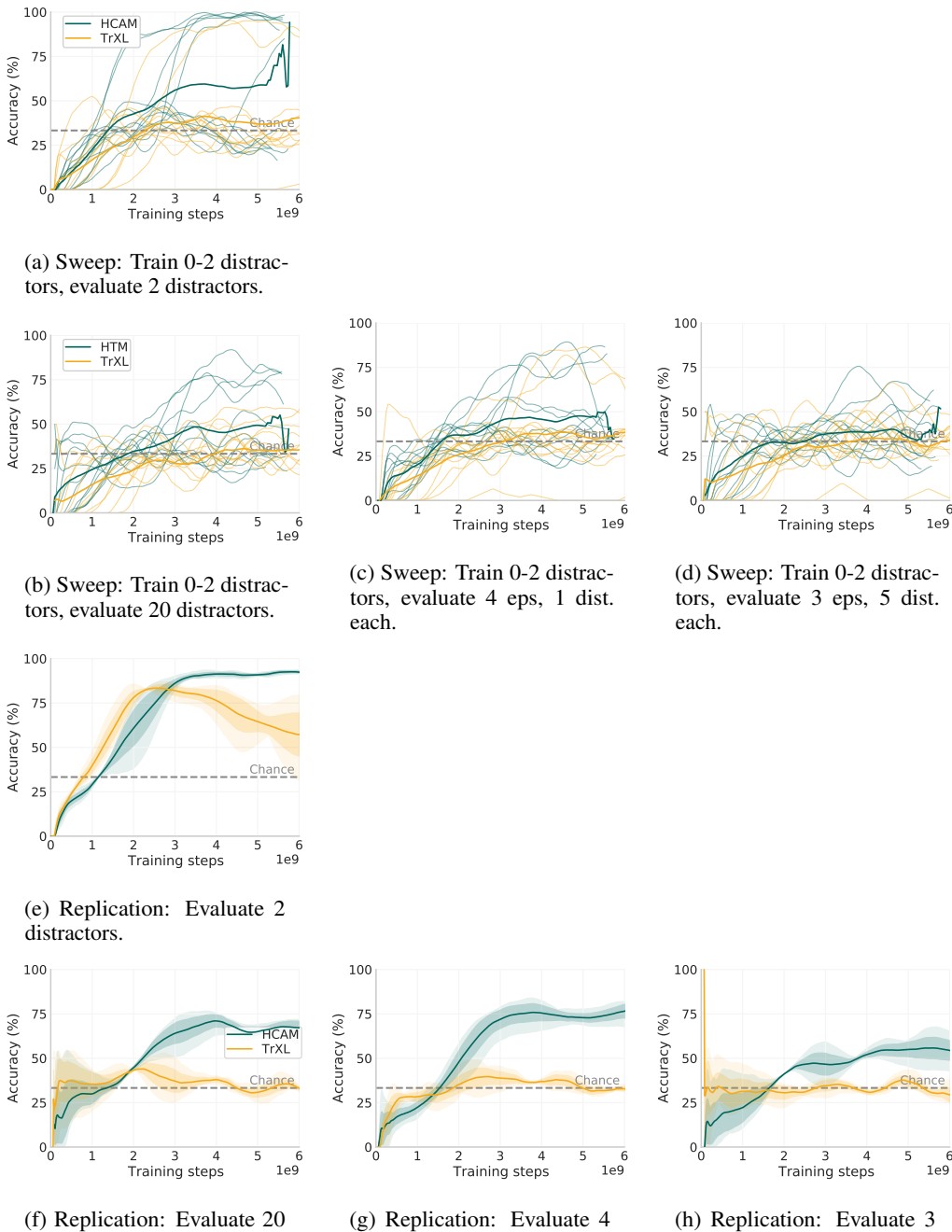

(a) Sweep: Train 0-2 distractors, evaluate 2 distractors.

(b) Sweep: Train 0-2 distractors, evaluate 20 distractors.

(c) Sweep: Train 0-2 distractors, evaluate 4 eps, 1 dist. each.

(d) Sweep: Train 0-2 distractors, evaluate 3 eps, 5 dist. each.

(e) Replication: Evaluate 2 distractors.

(f) Replication: Evaluate 20 distractors.

(g) Replication: Evaluate 4 eps, 1 dist. each.

(h) Replication: Evaluate 3 eps, 5 dist. each.

Figure 19: Sweeping hyperparameters (learning rate and entropy weight) in the Words tasks: HCAM is again more robust to variation in these hyperparameters. (a-d) The sweep results. The thick line plots the mean, while the thin lines plot individual sweep values. While HCAM is much more robust overall, as shown in the number of hyperparameter settings that learn the train tasks (a), a few TrXL jobs show above chance generalization on some tasks in the sweep. (e-h) To follow-up on the above result, we ran a replication of the best hyperparameters from the sweep with three new random seeds (as we did for all our main text results). This replication does not show substantially above-chance generalization performance from TrXL and shows some collapse in performance on the train tasks. HCAM's performance remains substantially above chance in a replication of the best values in the sweep, although note that the best results from the sweep are worse than the results with the tuned hyperparameters used in the original experiments. (Note also that these experiments were run with a longer attention window for HCAM than intended, see App. A.2.1.)