# OpenReview forum: "Towards mental time travel: a hierarchical memory for reinforcement learning agents"
_NeurIPS.cc/2021/Conference — NeurIPS 2021 Poster_

### Official Review · Reviewer_5ryt · 2021-07-15

**Rating:** 6
**Confidence:** 4

**Summary:**

The authors propose an architecture to augment an RL agent’s memory using a novel hierarchical transformer. The main novelties of the approach are 1) that the transformer attends over chunks and 2) that it’s used for long horizon RL. They test their approach in many interesting environments that each test a different component of memory (delayed recall, object permanence, distractors…). The results show that across the different environments the proposed approach outperforms the tested baselines. Moreover, the approach competes well with specially tuned state-of-the-art methods in other environments.

**Limitations And Societal Impact:**

No code and minimal plans to release – this will make it extremely hard for the community to iterate on the approach.


**Main Review:**


1 Strengths:

The main strength of the paper is the sheer number of different environments that were used to benchmark the method. It really is impressive and shows that the method can potentially handle different scenarios. Taking it a step further and comparing the architecture to state-of-the-art results in other domains is also a big plus.

Another strength is that the approach is fairly intuitive - attending over chunks instead of over individual states makes intuitive sense.

Many ablations are provided in the appendix.

2 Weaknesses:

Tuning the chunk size for each environment feels a little hacky. As mentioned by the authors – learning the chunk size would be an interesting avenue of future work.

3 Correctness:

From scanning the appendix – Figure 10 shows that the self-supervised loss seems to be crucial to obtain decent performance in the Ballet and Rapid word learning tasks. The model performs close to chance without it. I don’t think this is emphasized nearly enough in the main text. The method doesn’t work without it…

I find it hard to believe that the transformer XL performs so poorly on some of the simpler tasks. I can’t help but wonder if the baseline was properly tuned for these tasks. I scanned the appendix, and it seems like the same sweep over hyper parameters was performed for the main method as well as the baselines but I wonder if something as simple as tuning the learning rate for the baseline (which to my understanding was not done) could have improved its performance.

4 clarity:

For the most part the paper is well written and easy to follow. I think the paper could have benefited from a background section where Transformers and RL could have been introduced. As is, without any background, section 2 feels very dense. In particular, lines 95-109 are extremely dense and should be handled with more care.

I also think that Figure 11 from the appendix should somehow make its way into figure 5 of the main paper. It really feels strange to just show that the baselines are failing when in reality they do learn properly on the training task.

5 Relation to prior work:

The relation to prior work is handled nicely in the discussion section.

UPDATE: Thanks for the replies to my questions. I leave my score as it stands. (6)

**Time Spent Reviewing:**

3

---

> ### Author Response · Authors · 2021-08-09
> **(1/2) Thank you for your thorough review, and a summary of our response**
>
> Thank you for your thorough review—it’s great to see reviewers digging into the supplement and into the details of the method. We have run some new experiments specifically targeting learning rate and another hyperparameter, to address the related concerns that you and another reviewer raised, and have included the results below. We also appreciate your helpful suggestions about improving the communication of the results and methods in the paper, and we are incorporating them.
>
> Main response points:
> 1) Based on your suggestion, as well as another reviewer’s questions about how the baselines were tuned, we have clarified our communication about hyperparameter sources, and ran new experiments in two domains, sweeping learning rate and entropy cost (which we had not previously tuned, but had copied from previous papers), and found that our experimental conclusions did not change substantially with these new sweeps. In particular, HTM was more robust to variation in these hyperparameters, and TrXL did not learn the Ballet tasks better even with different learning rates.
>
> 2) We would like to note that our results seem relatively robust to chunk size, so it may not require substantial tuning. We reported this for Ballet in the supplement D.2, but have observed it in other domains as well, and could include demonstrations on other domains in the revised version if that would be useful.
>
> 3) We are in the process of open-sourcing an implementation of the hierarchical attention mechanism, as well as the Ballet and Rapid-Word-Learning environments, which will hopefully allow future researchers to reproduce our results and build off of our work.
>
> 4) We appreciate you pointing out a number of places where our paper could communicate more clearly, and are working on incorporating your suggestions.
>
> We believe these clarifications and new results show that our memory does substantially improve upon prior approaches, and we hope that they will address your concerns. We are incorporating these clarifications and new results into the paper (as well as clarifying/fixing other minor issues raised by you and other reviewers).

---

> > ### Author Response · Authors · 2021-08-09
> > **(2/2) Detailed responses**
> >
> > 1) New experiments on learning rate and entropy cost: In order to further address these concerns, we ran some new experiments exploring the effect of hyperparameters. In particular, because you suggested that different learning rates might improve the performance of TrXL, we swept the learning rate and entropy loss weight (which we had not varied previously from settings taken from prior papers) on both the Ballet and Rapid Word Learning tasks. We ran a full product of three learning rates that were above and below our original values (5e-4, 5e-5, 1e-5) and entropy costs that were 5x more or less than the original entropy cost, with 2 seeds per condition (for a total of 24 hyper x seed combination x memory type combinations per task domain). We emphasize that the same hyperparameters were tested for both models, and that these sweeps centered on hyperparameter settings that were previously untuned (sourced from prior papers), and that this sweep focuses on learning rate, which you suggested might benefit TrXL. We apologize for the length of this results description, but we wanted to communicate them fully and cannot provide a plot in this response.
> >     * Ballet results: HTM substantially outperforms TrXL in this sweep as well. First, HTM is far more robust to varying the hyperparameters—in every hyperparameter setting, the agents with HTM achieved off chance performance on the *hard 8-dance tasks* within 1 billion steps. By contrast, 58% of the TrXL jobs did not attain off chance performance on even the *easiest task* within 1.5 billion steps (when we stopped the training). In addition, 75% of the HTM jobs achieved above-chance performance (measured as window-averaged performance >5 percentage points above chance-level) on the easiest tasks before any of the TrXL jobs achieved above-chance performance on any task. The performance of the best TrXL jobs from this sweep looks similar to the performance at the same point in training from our original hyperparameters: about 40-50% performance on the 8 dance, short delays task, and 25-35% on the 8 dance, long delays task at 1.5 billion steps. HTM performed much better than TrXL, with 75% of the HTM jobs outperforming even the *best* TrXL agents on the hardest task, and the best HTM jobs comparable to the results in the paper, achieving 80-90% performance on the hardest tasks at 1.5 billion steps. In both 8 dance levels, the advantage of the two HTM seeds with the best hyperparams over the two TrXL seeds with the best hyperparams is significant by a paired* t-test, respectively t(1)=21, p=0.03 and t(1) = 101, p=0.006 (*paired reflecting the non-independence of the encoder initialization when the agents are initialized with the same random seed, but results are similar with an unpaired test, respectively t(2)=29, p=0.001; t(2)=9.5, p=0.01). In summary, TrXL’s performance at these tasks does not seem to be improved by varying learning rates or entropy cost, and HTM seems much more robust to variation in these hyperparameters.
> >     * Rapid word learning results: The results are similar. First, HTM is more robust to varying the hyperparameters: In these more challenging tasks, only 25% of the TrXL jobs achieve off-chance training performance within 5 billion steps, while 50% of the HTM jobs achieve above-chance performance on the training tasks. HTM also generalizes substantially better than TrXL. However, unlike our original experiments, one set of these TrXL jobs does achieve somewhat above-chance performance at the evaluation tasks we considered. Chance is 33.3%, and the best TrXL settings achieved 50% performance at the 20 distractors evaluation, 67% performance at the 4 episodes, 1 distractor phase test, and 42% performance at the 3 episodes, 5 distractors test. By comparison, the best HTM jobs from this sweep achieved 79%, 80%, and 59% generalization performance respectively (note that these jobs were not trained for as long as those in the paper). The advantage of the two HTM seeds with the best hyperparameters over the two TrXL seeds with the best hyperparameters was significant by a paired t-test in each case, respectively t(1)=28, p=0.02; t(1)=17, p=0.04; t(1)=56, p=0.01. (As a reminder, the HTM results using our original hyperparameters and training for longer achieved even more impressive performance: >95% on the first two evaluations, and 91% on the third.) Thus, HTM again appears to be both more robust across hyperparameters, and better when comparing best-hyperparameter configurations.
> >     * We will incorporate the results of both experiments into the paper, to show the better robustness and performance of HTM across a range of hyperparameters. We will also incorporate the fact that the TrXL memory does exhibit some non-chance generalization in the rapid word learning settings. Specifically, we will first replicate the improved TrXL results with the same number of independent seeds and training time used for the main experiments (as we did with all our experiments, since using the same seeds to select and report the results is biased), and incorporate those results.
> >     * Overall, we think that these experiments support our original claims that HTM is a substantial improvement over TrXL, and we hope that they help to assuage some concerns. If our paper is accepted, and there are other experiments of a similar scope that you think would help validate the conclusions for the final paper version, we would welcome your suggestions.
> >
> > 2) The performance of our architecture on the tasks from prior papers is also evidence that our benefits are not due to inadequate baseline tuning. In particular, although we did not highlight this clearly enough in the text, both the One-Shot StreetLearn and Paired Associative Inference papers compared models with some form of non-chunked attention, and found that they were unable to solve the task as effectively as HTM has. For example, our architecture matches the near-optimal performance of EPN on the One-Shot StreetLearn tasks. In particular, as we briefly noted in the text, the authors of the EPN paper evaluated agents with learned key-value episodic memories, but without chunking (Merlin and MRA) and found that they were unable to successfully learn to plan in One-Shot StreetLearn. Similarly, the original paper that proposed the Paired Associative Inference task evaluated a Universal Transformer, and found that it only achieved 84% performance, while HTM achieves 97.5% performance. Both of these results provide independent baseline comparisons that support the benefits of chunking over memory + attention alone.
> >
> > 3) We agree that choosing chunk sizes is limiting; however we also show in the appendix D.2 that our model’s performance is quite robust to different chunk sizes within a reasonable range, at least on the ballet tasks, so this parameter may not require extensive tuning. We observed this on other tasks while running hyperparameter sweeps, and have added a note about this in the supplement, and can add experimental demonstrations in other domains if you feel that would be useful. However, we still agree that learning to chunk optimally is a very interesting direction for future work—it could be that the same agent would benefit from different scales of chunking on memories from different experiences.
> >
> > 4) We agree with your suggestions for useful content to include in the main text: adding some further background on Transformers and RL, emphasizing further the contribution of the reconstruction loss (which was also emphasized in the prior work on rapid word learning), and showing training success from TrXL on the word-learning tasks. These are important points, which we did not have space to do full justice to in the paper. We should be able to fit these in the extra page of main text content if our paper is accepted.
> >
> > 5) Another reviewer pointed out that the acronym HTM is confusing given prior use in similar contexts. We have proposed some possible names below, and would appreciate any input you have on which convey the idea most clearly.
> >     * Hierarchical Attention to Chunks Memory (HAC Memory).
> >     * Hierarchical Chunk Attention Memory (HCAM).
> >     * Two-Level Transformer Memory (2LT or TLT)
> >     * Memory with Hierarchical Transformers (MHT)
> >
> > 6) We are working on preparing our hierarchical attention module and the ballet and rapid word learning tasks for an open-source release. We expect to be ready to release these in about a month, and will link them from the paper if it is accepted. Several of the other environments we used, such as One-Shot StreetLearn, have been released by the original authors, and so are available for comparison experiments. We will include links to the open source code for those as well in the relevant appendix sections, to facilitate comparisons.
> >
> > We hope that these experiments and responses will address your questions! Thank you for taking the time to consider them.

---

> ### Author Response · Authors · 2021-08-25
> **Thank you for your update**
>
> Thank you for taking the time to consider our response. We hope that this additional study confirms that the improvement over TrXL is clear in the settings we considered, but note that this hasn't changed your perception of the work substantially. Is there anything else we could do to convince you? For example, if it would be helpful for us to verify our progress on open sourcing the code and environments, we could present evidence to the SAC (as it would have to be de-anonymized), who could then confirm it to you.

---

### Official Review · Reviewer_3pZF · 2021-07-16

**Rating:** 8
**Confidence:** 4

**Summary:**

The authors proposed a memory network that breaks events into chunks, and retrieve memory hierarchically. They tested their model on a wide range of qualitatively different tasks and showed (sometime dramatic) improvement of performance against many baseline models

**Main Review:**

Originality
At the high-level, the idea of storing memories in a tree structure may not be mind-blowing. It’s still important to show that it can work in practice. And the range of tasks tested is rather impressive.

Quality
The quality of the work is high. The network architecture is intuitive and well-motivated. The empirical results are thorough and convincing.

Clarity
The paper is clearly written.

Significance
This work provides a solid and significant improvement in the use of memory network for RL agents.


**Time Spent Reviewing:**

1

---

> ### Author Response · Authors · 2021-08-09
> **Thank you for your thoughtful review, we appreciate your enthusiasm!**
>
> Thank you for your positive and encouraging review! We share your enthusiasm for the idea of more structured memory, we are glad you found the paper clear and compelling, and we appreciate your support for our work.
>
> We welcome any input on any of the comments from the other reviewers, for instance we are considering various possibilities for renaming the architecture based on the acronym conflict pointed out by another reviewer, and would appreciate any thoughts you have on which of the following conveys the idea most clearly:
> * Hierarchical Attention to Chunks Memory (HAC Memory).
> * Hierarchical Chunk Attention Memory (HCAM).
> * Two-Level Transformer Memory (2LT or TLT)
> * Memory with Hierarchical Transformers (MHT)

---

### Official Review · Reviewer_BE11 · 2021-07-17

**Rating:** 6
**Confidence:** 4

**Summary:**

The paper proposes a new RL agent architecture called Hierarchical Transformer Memory (HTM), which adds a history-chunking mechanism to the base GTrXL/TrXL model. HTM is found to perform well on six memory tasks.

**Limitations And Societal Impact:**

Yes

**Main Review:**

- Originality

The proposed architecture is novel. But the acronym HTM is too easily confused with the HTM network (Hierarchical Temporal Memory) introduced by Jeff Hawkins in 2004, and still in development today.


- Quality

The paper’s central claim is that HTM’s history-chunking mechanism improves its ability to retrieve information as needed. The most relevant experimental baseline is TrXL (essentially HTM without the chunking). HTM outperforms TrXL on four of six RL domains, but no TrXL results are reported for the other two tasks (Passive Visual Match and Paired Associative Inference). Nevertheless, HTM performs on par with SOTA models on those two tasks.

The case for HTM is undermined by the scarcity of hyperparameter tuning as reported. Reliable experimental practice generally requires that hyperparameters for baseline models be tuned with as much care as those of the new model. Table 1 (in Appendix A) lists the settings of 28 hyperparameters (HPs) for each of the 6 experiments. Four of those HPs pertain only to HTM, one pertains only to TrXL, and the other 23 apply to both models. The paper discusses the tuning of only 6 HPs (in Table 2 and Appendix D.3). How were the values of the other 22 HPs obtained? Most of them vary by task, and so were presumably found through limited testing of HTM on short runs instead of full sweeps. In no case are separate values reported, one for HTM and one for TrXL. Since TrXL is essentially an ablation of HTM, one would be tempted to tune many or most of the shared HPs on HTM, then apply the best HP configuration to TrXL as well. But this would inadvertently introduce a strong bias in favor of HTM over TrXL. Attention is the central transformer operation, and history chunking alters it in a major way, so there is little reason to believe that a given HP setting will prove equally good for HTM and TrXL.

All of the reported experiments were run for billions of training steps, demonstrating that sufficient compute was available to tune HPs for HTM and TrXL independently, if only on shorter runs. In the absence of separate tuning, it’s hard to assess the benefit of history chunking.


- Clarity

In general, the paper is very clear and well-written. Here are a few points for improvement:

Contrary to the answer on checklist item 3.b, section A.1 does not explain how most hyperparameters were chosen.

In Figures 3.a and 3.b, why does the HTM curve start at 100%?

In Figure 5, how could the LSTM do worse than random chance?


- Significance

As noted in “Limitations & future directions”, HTM requires the permanent storage of data from each step. The severity of this limitation strikes me as fundamental, and I don’t expect the potential workarounds to make it a practical approach for most applications of interest in the future. But this is a personal view, and I could be wrong.


- Conclusion

The open questions regarding the choices of hyperparameters lead me to doubt the attribution of HTM’s performance gains to its chunking mechanism. If I have overlooked or misunderstood something, I look forward to being corrected.

**POST-RESPONSE UPDATE**

I commend the authors for their helpful responses. They have clarified several issues related to hyperparameters, and have performed supplementary hyperparameter sweeps that improve the work.

Regarding the prior use of the term HTM, I find no problem with any of the suggested alternatives. HCAM seems the clearest to my mind.

I recommend replacing “chance level performance” with a clearer term, as it can too easily be interpreted to mean the performance of a trivial agent that simply takes random actions.

I have raised my score by one point.


**Time Spent Reviewing:**

12

---

> ### Author Response · Authors · 2021-08-09
> **(2/2) Detailed responses**
>
> 1) Thanks for pointing out the acronym conflict, we agree that HTM is confusing given prior use. We have proposed some possible names below, and would appreciate any input on which communicate the idea best.
>     * Hierarchical Attention to Chunks Memory (HAC Memory).
>     * Hierarchical Chunk Attention Memory (HCAM).
>     * Two-Level Transformer Memory (2LT or TLT)
>     * Memory with Hierarchical Transformers (MHT)
>
> 2) Thank you, we agree with the feedback on the hyperparameter section. We are revising the manuscript to specify hyperparameter sources clearly:
>     * Many hyperparameters that were not swept were taken from other sources without tuning. In particular, the first tasks we considered were the rapid word learning tasks, and many hyperparameters were taken directly from the original paper on which the tasks were based (“Grounded language learning: fast and slow”). These hyperparameters were therefore tuned by prior researchers for other models, including a TrXL memory, and so should not be biased towards our model. The visual encoder, language encoder, unsupervised reconstruction loss, entropy cost, etc. were copied from those described in the prior work.
>     * Since we ran all other experiments later, we used many of these same hyperparameters on other experiments, such as the visual and language encoder architecture across all experiments.
>     * Hyperparameters do sometimes differ across tasks (rather than across architectures) due to specific task features. E.g. the visual encoder for Ballet has a filter size of 9 because this is the resolution of each grid square, and the entropy cost for the ballet was chosen from a prior paper which used a similar action space These decisions were shared across all architectures, so should not favor HTM.
>
> 3) We are also revising the manuscript to clarify our experimental process.
>     * For TrXL, we indeed swept the hyperparameters in table 2 separately from HTM, but will describe this more clearly. The hyperparameters reported in the table are for HTM only. In our revised draft, we have included separate rows for TrXL for the swept hyperparameters.
>     * We note that we compared to a TrXL that was twice as wide or twice as deep as our main models (Appendix D.7), and found that it did not substantially change results, so this suggests that our results are also not due to width or number of layers being tuned for HTM.
>
> 4) To further address these concerns, we ran new experiments exploring the effect of hyperparameters. In particular, because a reviewer suggested that different learning rates might benefit TrXL, we swept the learning rate and entropy weight (which we had not varied previously from prior work) on both the Ballet and Word tasks. We ran a full product of three learning rates above/below our original values (5e-4, 5e-5, 1e-5) and entropy costs 5x more or less than the original value, with 2 seeds per condition (for a total of 24 hyper x seed combination x memory type combinations per task domain). We emphasize that the same hyperparameters were tested for both models, and that these sweeps centered on hyperparameter settings that were previously untuned (sourced from prior papers), and that this sweep focuses on learning rate, which a reviewer suggested might particularly benefit TrXL. We apologize for the length of this results description, but we wanted to communicate them fully and cannot provide a plot in this response.
>     * Ballet results: HTM substantially outperforms TrXL in this sweep as well. First, HTM is far more robust to varying the hyperparameters—in every hyperparameter setting, agents with HTM achieved off chance performance (measured as window-averaged performance >5 percentage points above chance-level) on the *hard 8-dance tasks* within 1 billion steps. By contrast, 58% of the TrXL jobs did not attain off chance performance on even the *easiest task* within 1.5 billion steps (when we stopped the training). In addition, 75% of the HTM jobs achieved above-chance performance  on the easiest tasks before any of the TrXL jobs achieved above-chance performance on any task. The performance of the best TrXL jobs from this sweep looks similar to the performance at the same point in training from our original experiments: about 40-50% performance on the 8 dance, short delays task, and 25-35% on the 8 dance, long delays task at 1.5 billion steps. HTM performed much better than TrXL, with 75% of the HTM jobs outperforming even the *best* TrXL agents on the hardest task, and the best HTM jobs comparable to the results in the paper, achieving 80-90% performance on the hardest tasks at 1.5 billion steps. In both 8 dance levels, the advantage of the two HTM seeds with the best hyperparams over the two TrXL seeds with the best hyperparams is significant by a paired* t-test, respectively t(1)=21, p=0.03 and t(1) = 101, p=0.006 (*paired reflecting the non-independence of the encoder initialization when the agents are initialized with the same random seed, but results are similar with an unpaired test, respectively t(2)=29, p=0.001; t(2)=9.5, p=0.01). In summary, TrXL’s performance at these tasks does not seem to be improved by varying learning rates or entropy cost, and HTM seems much more robust to variation in these hyperparameters.
>     * Rapid word learning results: The results are similar. First, HTM is more robust to varying hyperparameters: In these more challenging tasks, only 25% of the TrXL jobs achieve off-chance training performance within 5 billion steps, while 50% of the HTM jobs achieve above-chance performance on the training tasks. HTM also generalizes substantially better than TrXL. However, unlike our original experiments, one set of these TrXL jobs does achieve somewhat above-chance performance at the evaluation tasks we considered. Chance is 33.3%, and the best TrXL settings achieved 50% performance at the 20 distractors evaluation, 67% performance at the 4 episodes, 1 distractor phase test, and 42% performance at the 3 episodes, 5 distractors test. By comparison, the best HTM jobs from this sweep achieved 79%, 80%, and 59% generalization performance respectively (note that these jobs were not trained for as long as those in the paper). The advantage of the two HTM seeds with the best hyperparameters over the two TrXL seeds with the best hyperparameters was significant by a paired t-test in each case, respectively t(1)=28, p=0.02; t(1)=17, p=0.04; t(1)=56, p=0.01. (As a reminder, the HTM results using our original hyperparameters and training for longer achieved even more impressive performance: >95% on the first two evaluations, and 91% on the third.) Thus, HTM again appears to be both more robust across hyperparameters, and better when comparing best-hyperparameter configurations.
>     * We will incorporate the results of both experiments into the paper, to show the better robustness and performance of HTM across a range of hyperparameters. We will also incorporate the fact that the TrXL memory does exhibit some non-chance generalization in the rapid word learning settings. Specifically, we will first replicate the improved TrXL results with the same number of independent seeds and training time used for the main experiments (as we did with all our experiments, since using the same seeds to select and report the results is biased), and incorporate those results.
>     * Overall, we think that these experiments support our original claims that HTM is a substantial improvement over TrXL, and we hope that they help to assuage some concerns. If our paper is accepted, and there are other experiments of a similar scope that you think would help validate the conclusions for the final paper version, we would welcome your suggestions.
>
> 5) HTM's performance on existing tasks is also evidence that our benefits are not due to inadequate baseline tuning. We will highlight this more clearly in the text, but both the One-Shot StreetLearn and Paired Associative Inference papers compared models with some form of non-chunked attention, and found that they were unable to solve the task as effectively as HTM has. E.g. our architecture matches the near-optimal performance of EPN on OSSL. The authors of the EPN paper evaluated agents with learned key-value episodic memories, but without chunking (Merlin and MRA) and found that they were unable to successfully learn to plan in OSSL. Similarly, the paper that proposed the Paired Associative Inference task evaluated a Universal Transformer, and it only achieved 84% performance, while HTM achieves 97.5% performance. Both results provide independent baseline comparisons that support the benefits of chunking over memory + attention alone.
>
> 6) Thanks for pointing out the below-chance performance from some LSTM seeds in Fig. 5. Chance level performance is non-trivial: it requires picking up enough objects to get through all distractors, and then picking up one of the final objects. 2/3 LSTM seeds frequently failed to complete all distractor tasks before the episode timed out, and therefore had no chance at the final reward. Since the prior work showed that LSTMs failed to learn the simpler tasks above chance, we did not find this suspicious. But admittedly we ran the LSTMs on this and the object permanence task as a last-minute addition before submitting, so we did not use the full sweep that we used for the TrXL and HTM. (We did, however, sweep for the LSTMs on Ballet, where we expected them to be more competitive.) We have now swept these and the LSTM is able to more consistently achieve chance level performance. We will replicate these results with new seeds and update the figure.
>
> 7) Thanks for noting the curves starting at 100% as well. This is an artifact due to smoothing + very sparse data in this region being weighted heavily while the jobs are starting. We’ve now fixed this by dropping early data.
>
> We hope that these responses will address your questions! Thank you for taking the time to consider them.

---

> ### Author Response · Authors · 2021-08-09
> **(1/2) Thank you for your thoughtful review, a summary of our response**
>
> Thank you for your careful and thoughtful review—we appreciate the time you spent with our paper. We are also concerned about running fair experiments, and have done our best to keep comparisons fair while working on this project. You highlighted a number of important points where we hadn’t clearly communicated our experimental processes or the source of our hyperparameters, and identified a number of issues to fix. We have addressed these below, both by improving our communication and by running some new experiments to address the related concerns raised by both you and another reviewer.
>
> Summary (see other comment for more details on each point):
>
> 1) We agree that our acronym choice was confusing, and have proposed some new suggestions for your (and other reviewers’) feedback.
>
> 2) We didn’t communicate our hyperparameter sources and values clearly—the 22 values you mentioned largely came from prior papers or task constraints, not tuning, and we swept the indicated hyperparameters for TrXL and HTM separately, although we only reported the optimal values for HTM originally.
>
> 3) To further address these concerns, we ran new experiments in two domains, sweeping learning rate (as suggested by another reviewer) and entropy cost (which we had not previously tuned), and found that HTM was much more robust to these hyperparameters than TrXL, and that our experimental conclusions did not change substantially with these new sweeps.
>
> 4) We also emphasize that our comparisons on existing tasks, where in each case HTM achieved close to optimal or SOTA performance (while Transformers or key-value memories compared by the prior authors did not in 2 out of 3 cases), support our conclusion that our architecture is generally beneficial.
>
> We believe these clarifications and new results show that our memory does substantially improve upon prior approaches, and we hope that they will address your concerns. We are incorporating these clarifications and new results into the paper (as well as clarifying/fixing other minor issues raised by you and other reviewers).

---

### Decision · Program_Chairs · 2021-09-27

**Decision:**

Accept (Poster)

**Comment:**

- The proposed method is tackling an important problem, reasonable, and the experiments are well designed and executed.
- The major concerns from the reviewers (e.g., hyper parameter tuning, chunk size, TxXL results, etc.) are addressed well enough by the rebuttal.
- For reproducibility and accessibility, I suggest to open source the code and make the environment available at your earliest convenience.
- I suggest accepting the paper.